# Functional characterization of a multi-cancer risk locus on chr5p15.33 reveals regulation of *TERT* by ZNF148

Jun Fang[1,*], Jinping Jia[1,*], Matthew Makowski[1,2], Mai Xu[1], Zhaoming Wang[3,4], Tongwu Zhang[1], Jason W. Hoskins[1], Jiyeon Choi[1], Younghun Han[5], Mingfeng Zhang[1], Janelle Thomas[1], Michael Kovacs[1], Irene Collins[1], Marta Dzyadyk[1], Abbey Thompson[1], Maura O'Neill[6], Sudipto Das[6], Qi Lan[1], Roelof Koster[1], PanScan Consortium[†], TRICL Consortium[‡], GenoMEL Consortium[§], Rachael S. Stolzenberg-Solomon[3], Peter Kraft[7,8], Brian M. Wolpin[9,10], Pascal W.T.C. Jansen[2], Sara Olson[11], Katherine A. McGlynn[3], Peter A. Kanetsky[12], Nilanjan Chatterjee[3], Jennifer H. Barrett[13], Alison M. Dunning[14], John C. Taylor[13], Julia A. Newton-Bishop[13], D. Timothy Bishop[13], Thorkell Andresson[6], Gloria M. Petersen[15], Christopher I. Amos[5], Mark M. Iles[13], Katherine L. Nathanson[16], Maria Teresa Landi[3], Michiel Vermeulen[2], Kevin M. Brown[1,**] & Laufey T. Amundadottir[1,**]

Genome wide association studies (GWAS) have mapped multiple independent cancer susceptibility loci to chr5p15.33. Here, we show that fine-mapping of pancreatic and testicular cancer GWAS within one of these loci (Region 2 in *CLPTM1L*) focuses the signal to nine highly correlated SNPs. Of these, rs36115365-C associated with increased pancreatic and testicular but decreased lung cancer and melanoma risk, and exhibited preferred protein-binding and enhanced regulatory activity. Transcriptional gene silencing of this regulatory element repressed *TERT* expression in an allele-specific manner. Proteomic analysis identifies allele-preferred binding of Zinc finger protein 148 (ZNF148) to rs36115365-C, further supported by binding of purified recombinant ZNF148. Knockdown of *ZNF148* results in reduced *TERT* expression, telomerase activity and telomere length. Our results indicate that the association with chr5p15.33-Region 2 may be explained by rs36115365, a variant influencing *TERT* expression via ZNF148 in a manner consistent with elevated *TERT* in carriers of the C allele.

[1] Laboratory of Translational Genomics, Division of Cancer Epidemiology and Genetics, National Cancer Institute, National Institutes of Health, Bethesda, Maryland 20892, USA. [2] Department of Molecular Biology, Radboud Institute for Molecular Life Sciences, Radboud University, Nijmegen 6500 HB, The Netherlands. [3] Division of Cancer Epidemiology and Genetics, National Cancer Institute, National Institutes of Health, Bethesda, Maryland 20892, USA. [4] Cancer Genomics Research Laboratory, National Cancer Institute, Division of Cancer Epidemiology and Genetics, Leidos Biomedical Research, Inc., Frederick National Laboratory for Cancer Research, Frederick, Maryland 21702, USA. [5] Department of Community and Family Medicine, Geisel School of Medicine, Dartmouth College, Lebanon, New Hampshire 03756, USA. [6] Protein Characterization Laboratory, Cancer Research Technology Program, Frederick National Laboratory for Cancer Research, Frederick, Maryland 21701, USA. [7] Department of Epidemiology, Harvard School of Public Health, Boston, Massachusetts 02115, USA. [8] Department of Biostatistics, Harvard School of Public Health, Boston, Massachusetts 02115, USA. [9] Department of Medical Oncology, Dana-Farber Cancer Institute, Boston, Massachusetts 02215, USA. [10] Department of Medicine, Brigham and Women's Hospital and Harvard Medical School, Boston, Massachusetts 02215, USA. [11] Department of Epidemiology and Biostatistics, Memorial Sloan-Kettering Cancer Center, New York City, New York 10065, USA. [12] Department of Cancer Epidemiology, H. Lee Moffitt Cancer Center and Research Institute, Tampa, Florida 33612, USA. [13] Section of Epidemiology and Biostatistics, Leeds Institute of Cancer and Pathology, University of Leeds, Leeds LS9 7TF, UK. [14] Department of Oncology, University of Cambridge, Cambridge CB2 0XZ, UK. [15] Department of Health Sciences Research, Mayo Clinic, Rochester, Minnesota 55905, USA. [16] Translational Medicine and Human Genetics, Department of Medicine and Abramson Cancer Center, Perelman School of Medicine at the University of Pennsylvania, Philadelphia, Pennsylvania 19104, USA. * These authors contributed equally to this work. ** These authors jointly supervised the work. Correspondence and requests for materials should be addressed to K.M.B. (email: kevin.brown3@nih.gov) or to L.T.A. (email: amundadottirl@mail.nih.gov).
[†] The members of the PanScan Consortium are listed at the end of the paper. [‡] The members of the TRICL Consortium are listed at the end of the paper. [§] The members of the GenoMEL Consortium are listed at the end of the paper.

Risk variants across a small genomic region on chromosome 5p15.33 have been reported in genome wide association studies (GWAS) for at least eleven cancer types including bladder, breast, glioma, lung, melanoma, non-melanoma skin cancer, ovarian, pancreas, prostate, testicular germ cell cancer and chronic lymphocytic leukaemia[1–15]. Fine-mapping studies, either within a specific cancer type or across different cancers, have characterized up to seven independent loci in this region with either risk-enhancing or protective effects across a dozen cancers[16–18]. Notable is the fact that in nearly every locus, the effect is pleiotropic. This genomic region contains two plausible candidate genes, TERT and CLPTM1L. The former encodes the catalytic subunit of the telomerase reverse transcriptase (TERT), which in combination with an RNA template (TERC) adds nucleotide repeats to chromosome ends[19]. Although telomerase is active in germ cells and in early development, it is repressed in most adult tissues. Telomeres shorten with each cell division, and when they reach a critically short length, cellular senescence or apoptosis is triggered. However, cancer cells can continue to divide despite critically short telomeres, by upregulating telomerase or by alternative lengthening of telomeres (ALT) (refs 20–22). The CLPTM1L gene encodes the cleft lip and palate associated transmembrane 1-like protein, and is overexpressed in lung and pancreatic cancer where it promotes growth and survival and is required for KRAS driven lung cancer[23–27].

One of the multiple risk loci in this genomic region lies within the CLPTM1L gene and has been termed Region 2 (ref. 18), originally reported to be associated with risk of pancreatic, lung, bladder cancer, and melanoma, marked by either rs401681 or rs402710 (refs 1,4,5,11,28). By conducting fine-mapping across multiple cancers and subsequently investigating the functional consequences of the subset of genetic variants most strongly associated with cancer risk, we find that risk of pancreatic, testicular and lung cancer conferred by this locus may predominantly be explained by a single-SNP. This variant, rs36115365, exhibited preferred protein-binding and enhanced regulatory activity for the C-allele, associated with increased pancreatic and testicular but decreased lung cancer and melanoma risk.

Transcriptional gene silencing of the regulatory region encompassing this variant resulted in repression of TERT but not CLPTM1L expression in an allele-specific manner. Proteomic analysis identified allele-preferred binding of Zinc finger protein 148 (ZNF148) to rs36115365-C, a finding supported by binding of purified recombinant ZNF148 specifically to the C-allele, as well as by ChIP analysis showing allele-preferential binding of endogenous ZNF148 to rs36115365-C. Knockdown of ZNF148 resulted in reduced TERT expression, telomerase activity and telomere length. Taken together, these results indicate that the association with chr5p15.33-Region 2 may be explained by rs36115365, a variant influencing TERT via ZNF148 in a manner consistent with elevated TERT expression in carriers of the C allele.

## Results

**Fine-mapping the chr5p15.33 Region 2 risk locus**. We performed imputation and fine-mapping of the multi-cancer risk locus in the CLPTM1L gene (Region 2, originally marked by rs401681 and rs402710) using GWAS data for four cancers previously shown to have associations with this locus, namely pancreatic[11], testicular[28] and lung cancer[7], and melanoma[29]. For pancreatic cancer, fine-mapping identified SNPs with P values significantly lower than the previously published association signal marked by rs401681, with rs451360 being the smallest ($P = 2.0 \times 10^{-10}$ for rs451360; $P = 3.7 \times 10^{-7}$ for rs401681; Supplementary Table 1)[18]. This SNP is highly correlated with

eight other SNPs ($r^2 > 0.60$, 1000G EUR population) that collectively mark Region 2 in pancreatic cancer (Fig. 1). Fine-mapping of Region 2 for testicular germ cell tumours (TGCT) and lung cancer revealed that the strongest SNP for each was among this group of nine SNPs (rs35953391 for TGCT, $P = 1.08 \times 10^{-9}$; and rs37004 for lung cancer, $P = 1.18 \times 10^{-13}$; Supplementary Table 1). Conditional analysis for the most significant SNP across each cancer resulted in a substantial loss of the signal for the other eight SNPs in pancreatic ($P_{\text{Conditional}} = 0.47–0.91$), testicular ($P_{\text{Conditional}} = 0.21–0.92$) and lung cancer ($P_{\text{Conditional}} = 0.09–0.45$). In contrast, for melanoma none of the nine SNPs were significantly associated with risk in an unconditional analysis. However, upon conditioning on the most significant SNP in Region 2 (rs2447853, $P = 5.7 \times 10^{-12}$) (ref. 29), all nine SNPs became more significantly associated with melanoma risk ($P_{\text{Conditional}} = 5.77 \times 10^{-5}$ to $4.45 \times 10^{-3}$), consistent with the possibility that these SNPs may mark one or more risk variants independent of rs2447853.

We also noted in the 1000G Phase 3, version 1 reference dataset an insertion/deletion variant that was highly correlated with these nine SNPs (rs3030832, $r^2 = 0.96$ to rs451360 in EUR) that had not been included in the imputation reference based off an earlier version (1000G Phase 1, version 3). We therefore re-imputed the pancreatic cancer GWAS with the newer 1000G reference set and observed an association signal similar in strength and significance to that of the other nine variants (rs3030832, $P = 8.25 \times 10^{-10}$, OR = 1.28 95% CI 1.18–1.39; Supplementary Table 2) indicating that this indel variant should likewise be considered a candidate functional risk variant. Overall, these ten variants extend across the entire length of CLPTM1L, from the promoter to ~6 kb downstream of the gene (Fig. 1). Three variants, rs36115365, rs380145 and rs27071, are located within potential gene regulatory regions, annotated by the ENCODE project (Fig. 1, Supplementary Fig. 1).

**Allele-specific regulatory effects mediated by rs36115365**. We sought to assess whether any of the ten highly correlated sequence variants influence differential protein binding via electrophoretic mobility shift assays (EMSA) in the PANC-1 and/or MIA PaCa-2 pancreatic cancer cell lines (Fig. 2, Supplementary Fig. 2). Only rs36115365 exhibited allele-specific binding (Fig. 2), where the pancreatic cancer risk-associated minor C-allele (MAF 0.19 in 1000G EUR) displayed selective protein binding as indicated by greater loss of C-allele-specific banding upon addition of unlabelled C-allele competitor compared to unlabelled G-allele probe. EMSA assays for rs36115365 in seven additional cancer cell lines, including pancreatic cancer (MIA PaCa-2, Supplementary Fig. 3a), testicular germ cell cancer (NTERA-2 and 2102Ep, Supplementary Fig. 3b), lung cancer (A549, Fig. 2; NCI-H460, Supplementary Fig. 3c), and melanoma lines (UACC903 and UACC1113; Supplementary Fig. 3d) showed a similar pattern of allele-preferential binding to the C allele of rs36115365.

This SNP is located in-between the 5′ end of TERT (~18 kb upstream) and 3′ end of CLPTM1L (~5 kb downstream), a region that overlaps active histone modification marks and multiple transcription factor binding sites according to ENCODE data (Fig. 1, Supplementary Fig. 1). The region harbouring rs36115365 demonstrated an allele-specific increase in luciferase reporter activity as compared to empty vector that was consistent across all eight cancer cell lines tested (Fig. 3, Supplementary Fig. 4), including those from pancreas (PANC-1 and MIA PaCa-2, average fold change for C versus G allele 1.38, range 1.05–2.82), testis (NTERA-2, and 2102Ep, average fold change for C/G allele 1.95, range 1.12–4.83), lung (A549 and NCI-H460,

average fold change for C/G allele 1.33, range 1.05–1.95), and melanoma (UACC903 and UACC1113, average fold change for C/G allele 1.35, range 1.30–1.42). Transcriptional activity of the genomic region surrounding rs36115365 (240 bp) was higher in the forward (plasmids FG and FC) as compared to the reverse (plasmids RG and RC) orientation. Across all cancer cell lines, the C-allele on average showed an approximately 44% higher luciferase activity than the G-allele in the forward orientation, and 23% higher activity in the reverse orientation ($P = 4.2 \times 10^{-5}$–0.031).

Analysis of imputed GWAS data from pancreatic and testicular cancers conditioned on rs36115365 are consistent with rs36115365 accounting for the majority of the Region 2 signal ($P_{Conditional} = 0.03$–0.99 and $P_{Conditional} = 0.22$–0.92, respectively for the grouping of eight SNPs highly correlated with rs36115365; Supplementary Table 1), with the minor C allele being positively associated with risk. In lung cancer and melanoma, however, fine-mapping data suggest that the genetic architecture

underlying risk in Region 2 may be more complex, but are nonetheless consistent with a functional role for rs36115365. For lung, in contrast to pancreatic and testicular cancers, the C allele of rs36115365 is negatively associated with risk. Conditioning on rs36115365 revealed a possible secondary signal for lung cancer risk within the eight highly correlated SNPs ($P_{Conditional} = 3.74 \times 10^{-5}$–0.11; Supplementary Table 1). For melanoma, rs36115365 was not significant in single-SNP analysis ($P = 0.70$), but became more significant after conditioning on the best Region 2 SNP (rs2447853, $P_{Conditional} = 1.09 \times 10^{-4}$; Supplementary Table 1), with the C allele also being negatively associated with risk (OR = 0.86; 95% CI 0.80–0.93). After conditioning on rs36115365 for melanoma, rs2447853 also becomes more significant ($P_{Conditional} = 3.01 \times 10^{-15}$ versus $P = 5.7 \times 10^{-12}$). These data suggest rs36115365 may influence gene expression within the *TERT-CLPTM1L* region and may account for either some or the entire association signal in this region, depending on the cancer type.

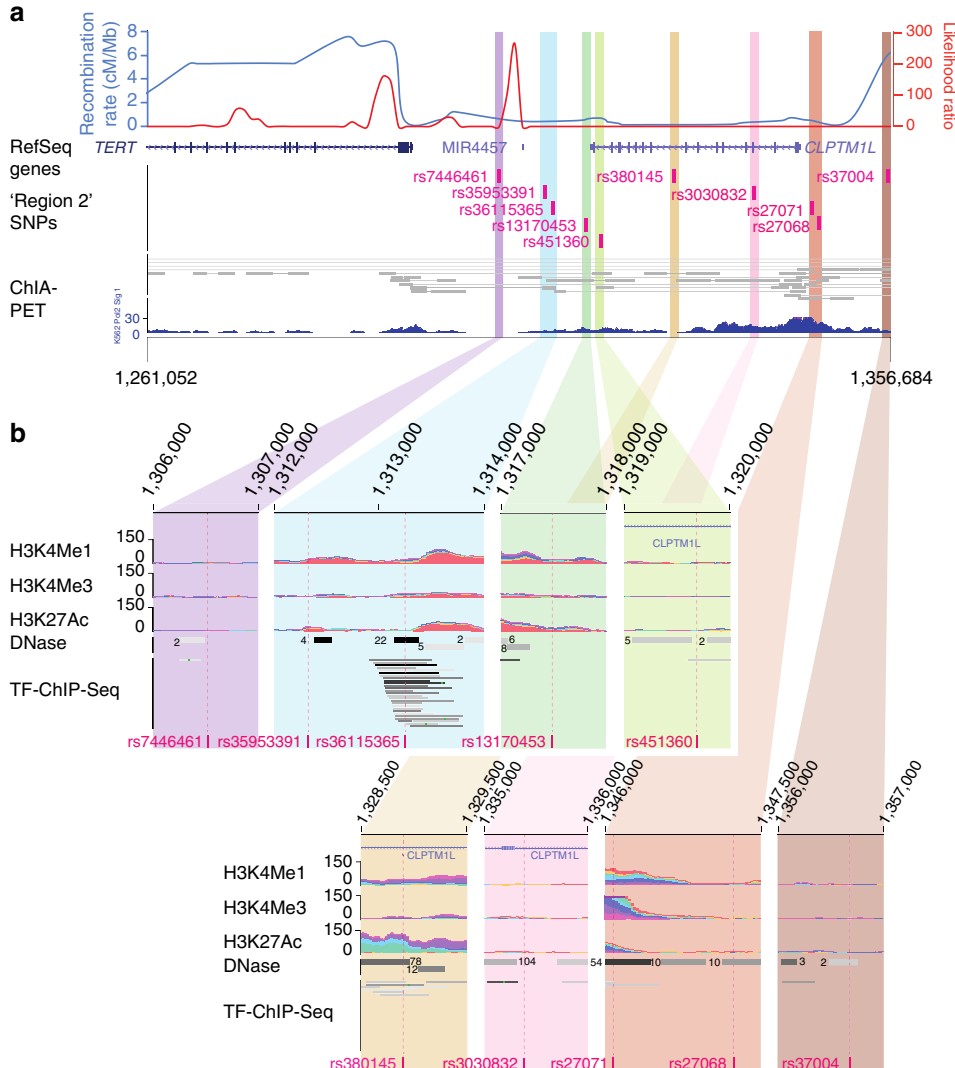

**Figure 1 | Map of *TERT* and *CLPTM1L* region.** Recombination hotspots in the CEU population (red line), as well as 1000G combined recombination rate (blue line) across the *TERT/CLPTM1L* region are shown relative to the *CLPTM1L* and *TERT* genes, as well as the grouping of ten highly correlated sequence variants strongly associated with risk of pancreatic, testicular, and lung cancers in the region closest to *CLPTM1L*. (**a**) Chromatin interaction analysis paired-end (ChIA-PET) sequencing data from the K562 chronic myeloid leukaemia cell line using an antibody against RNA polymerase II generated by the ENCODE project (https://www.encodeproject.org/) is shown. For each of the ten strongly associated variants, layered H3K4Me1, H3K4Me3, and H3K27Ac chromatin immunoprecipitation (ChIP-seq), DNAse I hypersensitivity sequencing (DNase) and transcription factor ChIP-seq (TF-ChIP-Seq) data from the ENCODE project are shown (**b**) as displayed by the UCSC Genome Browser (lower panels).

**Silencing the region harbouring rs36115365**. To interrogate whether the putative gene regulatory region harbouring rs36115365 influences expression of *TERT* and/or *CLPTM1L*, siRNA mediated transcriptional gene silencing (TGS) (refs 30,31) was used to target across this region to evaluate effects on gene expression. This mechanism of gene silencing is different from the well-known siRNA-mediated post-transcriptional gene silencing (PTGS) in that it targets a genomic regulatory region that mediates gene expression rather than messenger RNA (mRNA) (refs 30,31). Eight siRNAs were designed to span the region (Fig. 4a, Supplementary Table 3) and were separately transfected into cancer cell lines from pancreas (PANC-1), lung (A549), testis (NTERA-2), and melanoma (UACC903). Three of the eight (siRNA3, siRNA5 and siRNA8; Fig. 4b, Supplementary Fig. 5) showed significant inhibition of *TERT* mRNA expression by RT-qPCR in all four cell lines tested compared to a scrambled siRNA control, suggesting a role for the targeted region in the regulation of *TERT* expression. Inhibition of *TERT* by the three siRNAs ranged from 24 to 74% in PANC-1, 44 to 77% in A549, 33 to 49% in NTERA-2 and 54 to 84% in UACC903. The remaining five siRNAs showed little effect on expression of *TERT*. In contrast, expression of *CLPTM1L* as well as the *GAPDH* and *ACTB* housekeeping genes were not affected by any of the eight siRNAs. In addition, four siRNAs randomly designed to target non-genic regulatory regions on chromosome 8q24.21 were used as negative controls; none altered expression of *TERT*, *CLPTM1L*, or either housekeeping gene (Supplementary Fig. 6). The three siRNAs altered *TERT* expression in four additional cancer cell lines from pancreas (MIA PaCa-2), testis (2102Ep), lung (NCI-H460) and melanoma (UACC1113). Both siRNA3 and siRNA8 consistently reduced expression of *TERT*, but not *CLPTM1L* or housekeeping gene expression in all four lines, while siRNA5 resulted in specific down-regulation of *TERT* in some but not all lines (Supplementary Fig. 7). These data suggest that the genomic region harbouring rs36115365 plays a key role in the regulation of *TERT*, but not *CLPTM1L*, expression.

**Allele-specific *TERT* gene-regulatory activity by rs36115365**. We next sought to test whether the effect of TGS by siRNA targeting this putative gene-regulatory element on *TERT* expression was influenced by the genotype at rs36115365 by assessing allele-specific *TERT* mRNA expression. The human *TERT* gene harbours a synonymous SNP in exon 2 (rs2736098), linked to rs36115365 ($r^2 = 0.14$, $D' = 1.0$ in 1000G CEU), allowing for assessment of expression of *TERT* from chromosomes harbouring the C and G alleles of rs36115365 in cell lines heterozygous for both SNPs. We screened genomic DNA and complementary DNA (cDNA) from 55 pancreatic cell lines, as well as the melanoma, lung and testis cancer cell lines from the NCI60 panel to identify cell lines that are both heterozygous for rs36115365 and express two different alleles of rs2736098, yielding two assayable pancreatic cancer cell lines (Panc 05.04, IMIM-PC-1) and one lung cancer cell line (A549). The two pancreatic cancer cell lines express higher levels of *TERT* from the C as compared to the G allele (2.3 and 9.8 fold, respectively) whereas A549 cells express higher levels from the G allele (Supplementary Fig. 8). However, after adjusting for DNA copy number, all three cell lines express higher levels of *TERT* from the C allele (1.2 fold for A549 cells). We evaluated allele-specific levels of inhibition of *TERT* expression by siRNA3 (which is both closest to rs36115365 and most consistently inhibits *TERT* expression across the cell lines previously tested) in these three cell lines using a TaqMan allelic-discrimination assay for rs2736098. Inhibition by the siRNA on the C versus the G allele of rs36115365 was 60.2 versus 49.1% in Panc 05.04 cells ($P = 0.007$; t-test), 70.0 versus 63.6% in A549 cells ($P = 0.003$; t-test) and 28.3 versus 16.4% in IMIM-PC-1 cells ($P = 0.002$; t-test) (Fig. 4c).

These results indicate that rs36115365 lies in a gene-regulatory element that influences *TERT* expression in an allele-specific manner.

**Zinc-finger transcription factor 148 binds rs36115365-C**. To investigate the underlying mechanism of the differential gene regulation by genotypes at rs36115365, and to identify transcription factors potentially mediating this effect, we performed pull-down with oligonucleotides corresponding to the C or the G allele of rs36115365 incubated with nuclear extracts from PANC-1 and UACC903 cell lines, followed by quantitative mass spectrometry[32]. While most proteins identified bind both variants equally well, we noted outliers that bound the C-allele preferentially over the G-allele, as demonstrated by their location on the two-dimensional interaction plot (lower left quadrant of each, Fig. 5a and Supplementary Fig. 9), consistent with the EMSA data and suggesting preferential protein binding to this allele. Three proteins (ZNF148, VEZF1/ZNF161 and ZNF281) were identified as binding the C variant of rs36115365 preferentially in label-swapping experiments performed across both PANC-1 and UACC903 cell lines using a poly-dAdT competitor (Fig. 5a). A fourth protein, ZNF740, was also found to preferentially bind the C variant in both cell lines using mixed poly-dAdT and poly-dIdC competitors (Supplementary Fig. 9, bottom panels). We sought to verify whether any of these four proteins differentially bound the C-allele by using antibodies against these proteins in conjunction with EMSAs for rs36115365 (Fig. 5b, Supplementary Figs 10 and 11). Only the antibody against ZNF148 consistently resulted in loss of C allele-specific banding in pancreatic (PANC-1; Fig. 5b), as well as testis (NTERA-2) and lung cancer (A549) lines (Supplementary Fig. 10). Furthermore, EMSAs using recombinant purified ZNF148 protein demonstrated specific binding of ZNF148 to the C allele of rs36115365 (Fig. 5c). Notably, the resulting band had similar mobility characteristics to both those from EMSAs of ZNF148 bound to a known binding site in the *CDKN1A/p21* promoter[33,34], as well as the C allele-specific band for rs36115365 using PANC-1 nuclear extracts (Fig. 5c). Consistent with these data, ZNF148, (also named ZBP-89) a zinc-finger transcription

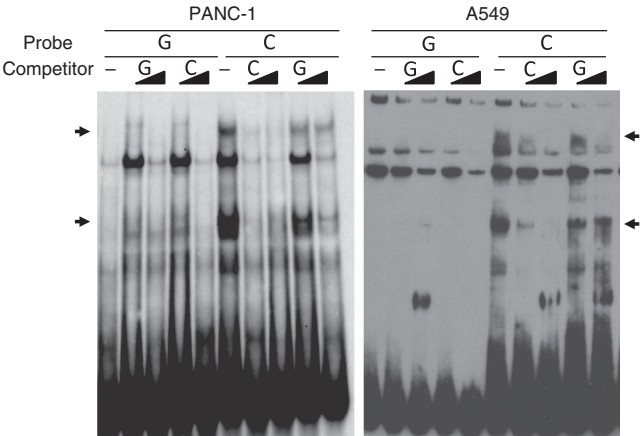

**Figure 2 | rs36115365 preferentially binds a nuclear protein.**
Electrophoretic mobility shift assays (EMSA) with biotin-labelled oligonucleotides containing either rs36115365-C or rs36115365-G in pancreatic (PANC-1) and lung (A549) cancer cell line nuclear extracts. Two specific protein complexes bind the C allele of rs36115365 preferentially in both cell lines and are more strongly competed with unlabeled C probe as compared to unlabeled G probe. Unlabelled competitor was used at ×10 and ×100 (as indicated by gradient symbol). Arrows denote specific protein complexes bound by the C allele of rs36115365.

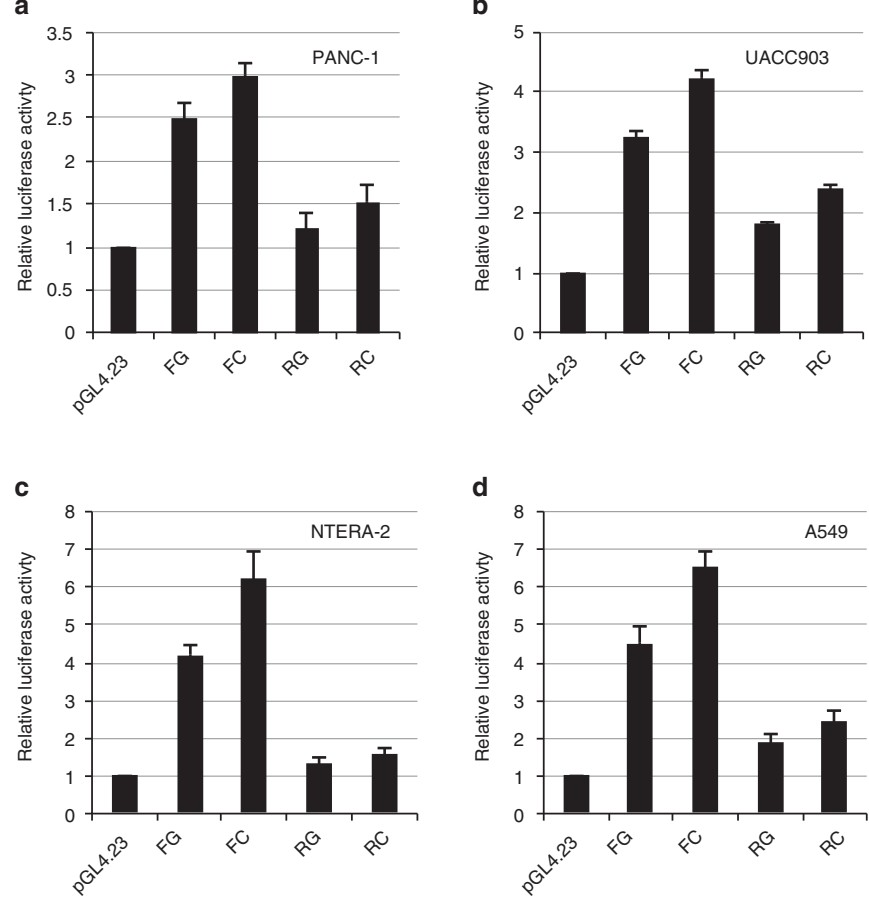

**Figure 3 | Regulatory activity for the genomic region containing rs36115365.** (**a**) Allele-specific luciferase activity was tested for rs36115365 in four cancer derived cell lines from pancreas (PANC-1) (**b**) melanoma (UACC903) (**c**) testis (NTERA-2) and (**d**) lung (A549). DNA fragments containing either the C or G allele at rs36115365 were cloned upstream of a minimal promoter (pGL4.23) driven reporter in the forward (F) or reverse (R) genomic orientation followed by transient transfections and luciferase assays. Luciferase activity was normalized to empty vector. Experiments were performed in triplicate and repeated at least three times; errors bars represent the standard error of the mean (s.e.m.). Supplementary Fig. 4 contains luciferase results for one additional cell line per tumour type (MIA PaCa-2, UACC1113, NTERA-2, 2102Ep and NCI-H460).

factor of the kruppel-like family[35], is predicted to bind to a consensus DNA-recognition motif created by the C-allele of rs36115365 (Fig. 5d). To further establish the binding of ZNF148 to rs36115365 and surrounding genomic region, we performed chromatin-immunoprecipitation (ChIP) for ZNF148 followed by quantitative PCR, noting an enrichment of binding at rs36115365 in pancreatic and lung cancer cell lines homozygous and heterozygous for rs36115365-C as compared to background and the surrounding area (Fig. 5e, Supplementary Fig. 12a–f). We also assessed allelic enrichment in the immunoprecipitates and noted a significant enrichment of the C allele as compared to the G allele in A549 cells (1.51 fold, $P = 0.01$; t-test; Supplementary Fig. 12g), with Panc 05.04 cells showing a nonsignificant trend in the same direction (1.12 fold, $P = 0.06$; t-test; Supplementary Fig. 12g).

**ZNF148 knockdown reduces TERT mRNA and telomerase activity.** To determine the effect of ZNF148 depletion on expression of TERT and CLPTM1L in pancreatic, lung, testicular, and melanoma cell lines ($n = 8$ total), we used siRNA-mediated PTGS. We observed that while depletion of ZNF148 resulted in little change in expression of CLPTM1L, expression of TERT was significantly decreased in most of the cell lines, with an average expression of 0.50 relative to a scrambled siRNA control (range 0.27–0.89, $P = 2.0 \times 10^{-4}$–0.012; t-test; Fig. 6a, Supplementary Figs 13 and 14), consistent with a role for ZNF148 in regulating

TERT expression. In contrast, siRNA-mediated knockdown of VEZF1 (ZNF161), ZNF281 and ZNF740 showed no effect on expression of either TERT or CLPTM1L (Supplementary Fig. 15). We next sought to assess if ZNF148-mediated regulation of TERT expression was accompanied by effects on telomerase activity and telomere length. Knockdown of ZNF148 via PTGS resulted in reduced telomerase activity in A549 and MIA PaCa-2 cells (Fig. 6b), as well as in NTERA-2 and UACC903 cells (Supplementary Fig. 16). This reduction was similar to that observed via siRNA-mediated depletion of TERT itself, or by transcriptional gene silencing (TGS, siRNA3) to target the gene regulatory element encompassing rs36115365. To further assess the role of ZNF148 in regulating TERT expression and activity, we performed rescue experiments after depletion of endogenous ZNF148 using an siRNA targeting the 3′-UTR of ZNF148. Overexpression of exogenous ZNF148 lacking the 3′-UTR indeed rescued both TERT expression and telomerase activity in A549 and MIA PaCa-2 cells (Supplementary Fig. 17). Consistent with these data, depletion of either ZNF148 or TERT, or alternatively targeting the rs36115365 regulatory region in both A549 and MIA PaCa-2 cells all resulted in similar reductions of telomere length (Fig. 6c).

## Discussion

A small genomic region on chr5p15.33, that harbours the TERT and CLPTM1L genes, has been reported to influence risk of

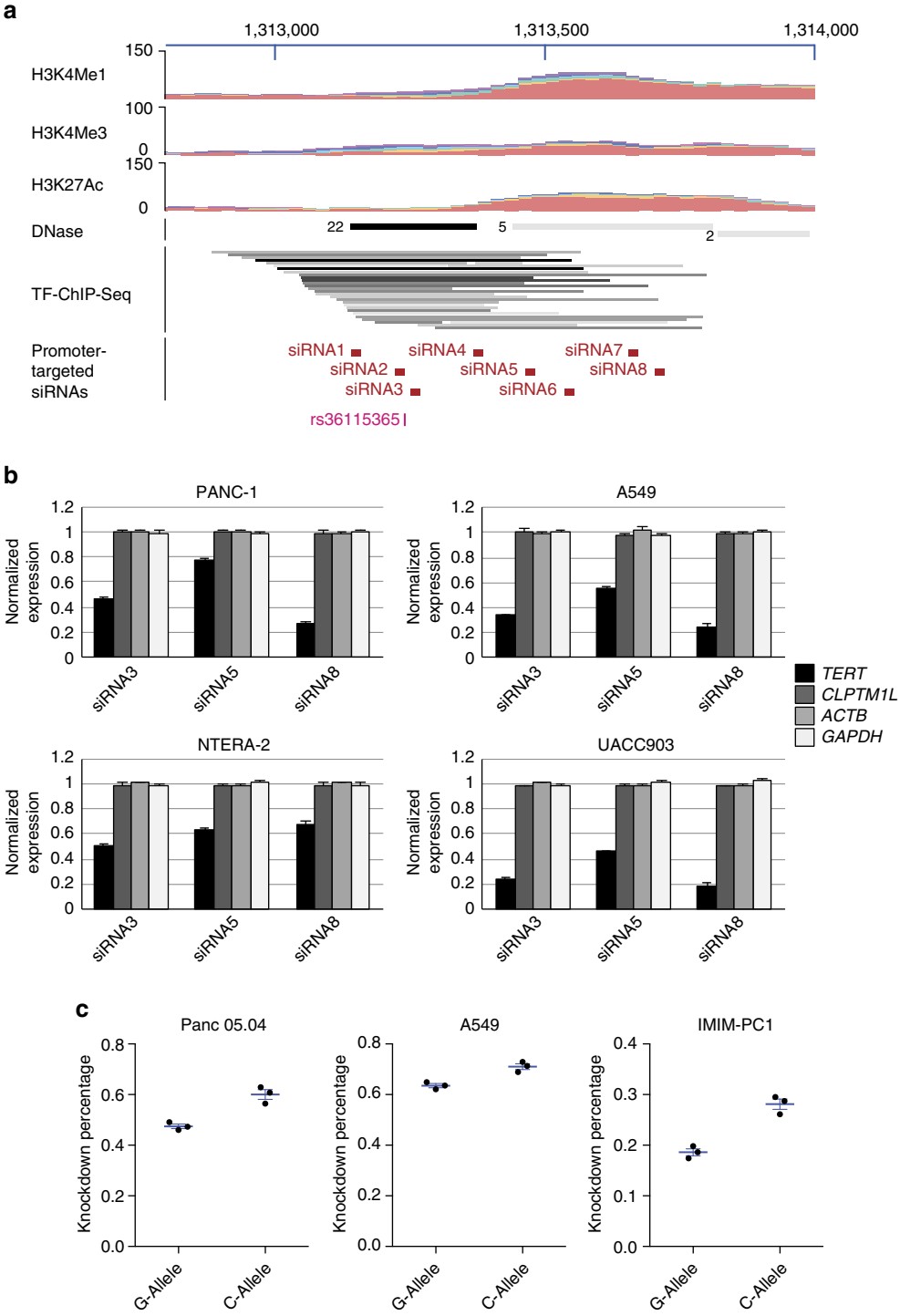

**Figure 4 | siRNA targeting the region near rs36115365 reduces expression of *TERT*.** (**a**) Eight double-stranded siRNAs were designed to target the potential gene-regulatory region encompassing rs36115365. (**b**) Three of eight double-stranded siRNAs targeting the rs36115365 locus can inhibit the gene expression of *TERT* but not *CLPTM1L*, *ACTB*, and *GAPDH* when transfected into PANC-1 (pancreatic cancer, upper left), A549 (lung cancer, upper right), NTERA-2 (testicular cancer, lower left) and UACC903 (melanoma, lower right). Expression values were normalized to a scrambled control siRNA. Experiments were performed in triplicate and repeated at least three times; error bars represent standard error of the mean (s.e.m.). (**c**) Allele-specific *TERT* expression after siRNA targeting (by siRNA3) of the regulatory region encompassing rs36115365. *TERT* expression was assayed using a quantitative allelic-discrimination TaqMan assay for a synonymous mRNA coding SNP in the *TERT* gene (rs2736098) that is genetically linked to rs36115365 ($r^2 = 0.14$, $D' = 1.0$). Two pancreatic cancer cell lines (Panc 05.04, left; and IMIM-PC-1, right) and one lung cancer line (A549, center) heterozygous for both SNPs were assayed. Expression of the allele of rs2736098 that is linked to the C-allele of rs36115365 (labelled 'C allele') was reduced to a greater extent than the allele linked to rs36115365-G. The degree of knockdown of each allele was normalized to that from a scrambled siRNA control. Experiments were performed in triplicate and repeated three times. Mean measures for three independent experiments are plotted; error bars represent s.e.m.

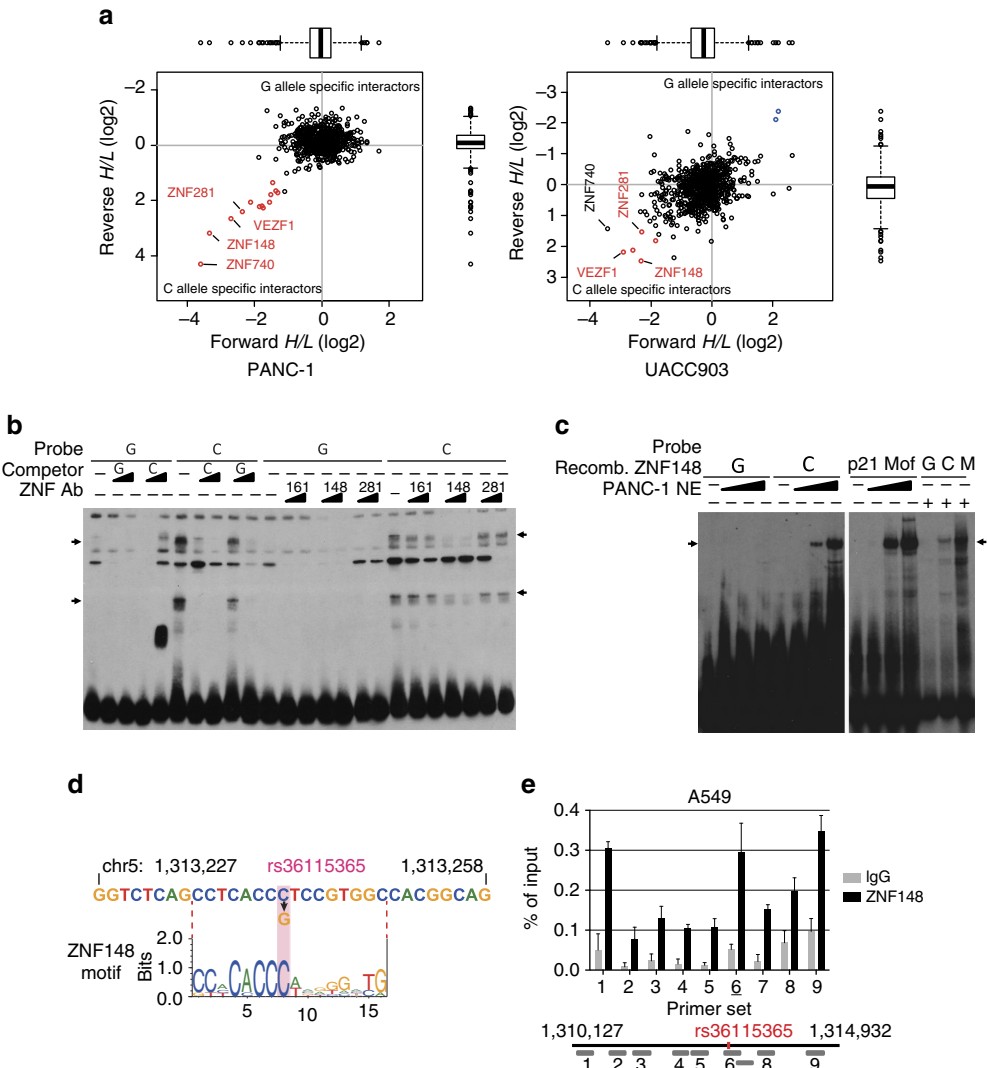

**Figure 5 | ZNF148 is an allele-specific interactor of rs36115365.** (**a**) DNA pulldowns were performed using PANC-1 (pancreatic cancer) and UACC903 (melanoma) cell line nuclear extracts with 41 bp biotin-tagged rs36115365 bait oligos. Replicate label-swapping experiments were performed with poly-dAdT competitor. Ratios indicate enrichment for protein binding to either the C- or G-allele. Significantly enriched ratios in both a forward and reverse label-swapping experiment were called as outliers (red: C-allele; blue: G-allele. Outlier cutoff is 1.5 IQRs). Identities of C-allele preferential binding proteins identified as outliers. Boxes in boxplots represent first to third quartiles and whiskers extend to furthest data point still within 1.5 IQRs of either quartile. (**b**) EMSA was performed with oligonucleotides containing the C or G allele at rs36115365 using nuclear extract from PANC-1 pancreatic cancer cells. The left side (lanes 1–10) shows preferred binding of two protein complexes to the C allele of rs36115365; the right side (lanes 11–24) competition of binding to the C allele by antibodies specific for three zinc finger proteins: VEZF1/ZNF161, ZNF148/ZBP-89 and ZNF281. The gradient symbol indicates 1 or 2 µg antibody used for competition. An antibody to ZNF148/ZBP-89 competes efficiently for binding of a protein to the rs36115365-C oligonucleotide. (**c**) EMSA showing binding of recombinant human ZNF148 to the C allele of rs36115365 (lanes 1–8). A ZNF148 binding site in the *CDKN1A/p21* promoter (p21 Motif, M) was used as a positive control (lanes 9–12), as well as PANC-1 nuclear extract (lanes 13–15). Arrows denote proteins bound by the C allele of rs36115365. The gradient symbol indicates addition of 90, 360 or 630 ng of recombinant ZNF148 to the reaction. (**d**) rs36115365 alters a predicted ZNF148/CACCC-binding protein DNA recognition motif (Transfac motif M00721). (**e**) Chromatin immunoprecipitation (ChIP) using an antibody against ZNF148 (black bars) in the A549 lung cancer cell line revealed enrichment of ZNF148 binding at rs36115365 (amplicon #6) over background levels using a nonspecific IgG antibody (grey bars). The location of amplicons tested with qPCR relative to rs36115365 are shown underneath the graph (drawn to scale). PCRs were conducted in triplicate and repeated for each of two immunoprecipitations; error bars represent standard deviation (s.d.) for a representative experiment.

multiple cancers and may contain up to seven or more independent susceptibility loci[16–18]. The complexity of this locus is highlighted by the fact that the same alleles confer susceptibility to some cancers while they are protective for others. One of these susceptibility loci termed Region 2, initially marked by rs401681 and rs402710 in *CLPTM1L*, was fine-mapped in a subset-based meta-analysis across multiple cancer types[18] and is the focus of the current study. The ten variants that mark Region

2 span the whole length of *CLPTM1L* to ~17 kb upstream of the transcriptional start site of *TERT*. Here, we identify rs36115365 as a functional SNP in this region and provide a plausible biological explanation underlying risk, featuring altered *TERT*, but not *CLPTM1L*, expression. Fine-mapping of Region 2 using GWAS data from pancreatic, lung and testicular cancer confirmed significant association with this small set of tightly linked SNPs (Fig. 1). Little signal remained within Region 2 after accounting

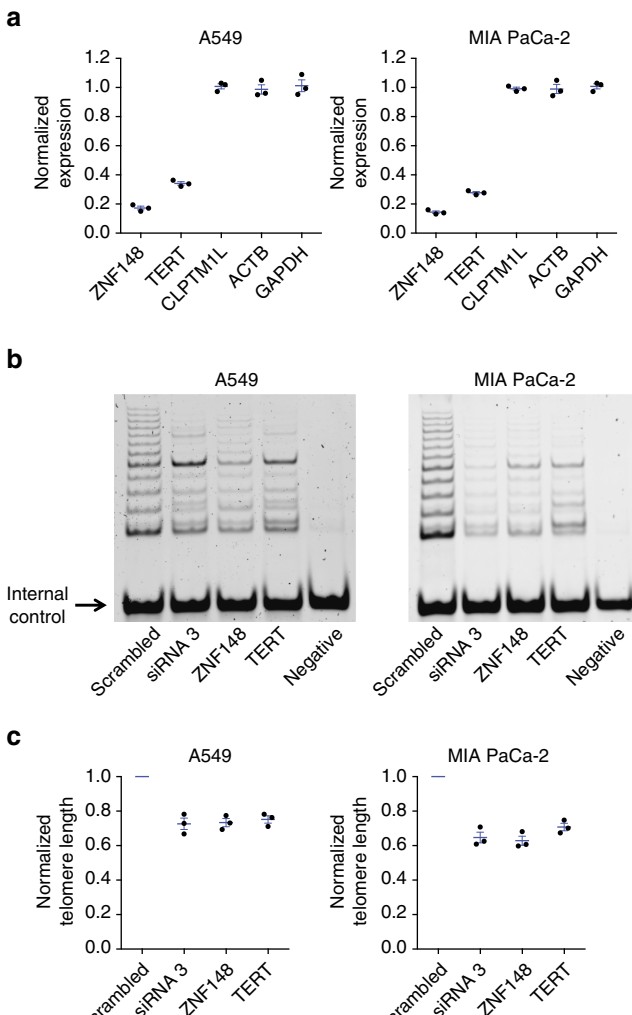

**Figure 6 | *ZNF148* depletion results in decreased *TERT* expression and telomerase activity.** (**a**) An siRNA directed against the *ZNF148* transcript was transfected into A549 (lung, left) and MIA PaCa-2 (pancreas, right) cell lines, and expression of *ZNF148*, *TERT*, *CLPTM1L*, *ACTB* and *GAPDH* were assayed by quantitative PCR. Depletion of *ZNF148* resulted in consistent reduction of *TERT* but not *CLPTM1L* or control gene expression. Expression values were normalized to those from cells transfected with a scrambled control siRNA. Experiments were conducted in triplicate and repeated three times. Mean measures for three independent experiments are plotted; error bars represent s.e.m. (**b**) siRNAs targeting *ZNF148*, *TERT*, the regulatory region encompassing rs36115365 (siRNA3), or a scrambled siRNA control were transfected into A549 (lung, left) and MIA PaCa-2 (pancreas, right) cell lines, and telomerase activity was measured via a telomeric repeat amplification protocol (TRAP). Negative control represents the TRAP assay performed with no cell extracts added. The internal control represents the 36 bp internal standard. (**c**) siRNAs targeting *ZNF148*, *TERT*, or the regulatory region encompassing rs36115365 (siRNA3) were transfected into A549 (lung, left) and MIA PaCa-2 (pancreas, right) cell lines repeatedly (once every four days), and telomere length was measured after 20 days using quantitative PCR for telomere repeat copy number. Data are normalized to those from a scrambled siRNA control. Experiments were conducted in triplicate and repeated three times. Mean measures for three independent experiments are plotted; error bars represent s.e.m.

for rs36115365 or alternatively the respective most significant SNP in pancreatic and testicular cancer, consistent with the notion that one or more of these variants (and/or an as-of-yet unidentified variant tightly linked with these SNPs) is responsible

for mediating cancer risk attributable to this locus. For lung cancer, residual signal was seen after conditional analysis on rs36115365 ($P_{Conditional} = 3.74 \times 10^{-5}$ to $7.71 \times 10^{-4}$), indicating that this SNP may not explain the entire Region 2 signal for lung cancer. For melanoma, a SNP (rs2447853) highly correlated to the original GWAS SNP reported for these cancers (rs401681, $r^2 = 0.97$) represents the most significant SNP in Region 2 (ref. 29). Although rs36115365 was non-significant in single-SNP analyses, it became more significant after conditioning on rs2447853 ($P_{Conditional} = 1.09 \times 10^{-4}$). The LD structure between these SNPs and conditional analyses suggest that in melanoma both may mark independent functional variants, with the signal at rs2447853 masking the association between rs36115365 and melanoma risk in single-SNP analysis.

In contrast with the other variants, preferred protein binding was seen on the minor (C) allele of rs36115365 across multiple cell lines representing all four cancer types. Luciferase reporter assays consistently showed differential gene regulatory activity between alleles across the cancer cell lines assayed. These data suggested rs36115365 as a strong candidate for a functional multi-cancer risk variant but did not specifically implicate which gene(s) may be influenced by this SNP.

While a suite of tools is commonly used to interrogate potential gene-regulatory GWAS loci and link regulatory variants to a specific gene or genes[36], their application was challenging for this locus. Expression quantitative trait locus analysis proved problematic for *TERT* given the relatively low expression of this gene in normal tissues. Likewise, the utility of chromosome conformation capture (3C) methods to establish a physical association between risk variants and specific target genes is greatly limited by the relatively short distances between rs36115365 and the *TERT* promoter. To establish a relationship between this element and regulation of gene expression, we targeted the intergenic risk region using siRNAs. This method has previously been used to inhibit promoter function via small RNA duplexes by a process termed TGS (refs 30,31,37–41). We applied this methodology to our study of an intergenic GWAS susceptibility variant, and established a role for the regulatory element in driving *TERT* (but not *CLPTM1L*) gene expression across multiple cancer types. These data suggest that the method may be of broader utility in the functional interrogation of GWAS loci.

Our results suggest that the binding of one or more proteins to the C-allele of rs36115365 is likely to play a key role in regulating *TERT* expression. Through quantitative mass spectrometry, we identified preferential binding of zinc finger protein 148 (ZNF148, also named ZBP-89) to the C-allele of rs36115365 in multiple cancer cell lines, and ChIP data confirmed binding of ZNF148 over rs36115365. We observed a subtle but significant preference for ZNF148 binding to the C-allele in ChIP experiments using A549 lung cancer cells, with a non-significant trend in the same direction in Panc 05.04 pancreatic cancer cells. These subtle differences in transcription factor binding are consistent with the very small effects this locus confers on cancer risk over a person's lifetime. Consistent with a central role for ZNF148 in regulating expression of *TERT*, siRNA-mediated gene knockdown of *ZNF148* consistently resulted in reduced expression of *TERT*. Furthermore, both knockdown of ZNF148 as well as TGS of the gene regulatory element in which rs36115365 resides reduced telomerase activity and telomere length, to a degree similar to knockdown of *TERT* itself. After depletion of ZNF148, this effect was rescued by exogenous ZNF148.

ZNF148 is a transcriptional regulator of the kruppel-like family that binds GC-rich DNA sequences in a variety of promoters to either activate or repress gene expression (reviewed by Zhang *et al.*[42]). Overexpression of ZNF148 promotes growth arrest and

apoptosis in gastrointestinal cancer cell lines *in vitro* and suppresses adenoma formation in the ApcMin/+ mouse model *in vivo*[43]. This may be due, at least in part, to ZNF148 binding of p53 that prevents nuclear export and results in elevated levels of nuclear p53 (ref. 44). ZNF148 has also been shown to be important in regulating *CDKN1A* gene expression, binding a GC-rich element in the promoter of the gene and recruiting both ataxia-telangiectasia mutated kinase and histone acetyltransferase p300 into a complex that drives histone deacetylase inhibitor (HDACi) mediated induction of this gene[33,34].

Our results indicate that ZNF148 may regulate *TERT* expression in pancreatic, testicular, lung, and melanoma tumour cell lines via a regulatory element that is disrupted by the G allele at rs36115365. As some of these cell lines have *TERT* promoter mutations (UACC903, UACC1103) whereas others do not (PANC-1, MIA PaCa-2, unpublished data), our results indicate that regulation by ZNF148 is important even in the presence of these presumably activating mutations.

In summary, our work has uncovered a likely causal variant in the *TERT-CLPTM1L* Region 2 susceptibility locus and identified ZNF148 as a potential effector of a gene-regulatory element that mediates increased *TERT* expression in an allele-specific manner. Furthermore, our fine-mapping results highlight the complexity of this region and indicate that Region 2 may, in some cancers, consist of more than one underlying functional signal. Our results are remarkably consistent in eight cell lines across four different cancer types and explain, at least in part, the biological underpinnings of risk for rs36115365. Notably, our data suggest that the mechanism by which ZNF148 influences *TERT* is similar for cancer types in which the C-allele of rs36115365 contributes to increased risk, or alternatively to disease protection. Although *TERT* expression and ensuing effects on telomere length may be the crucial underlying mechanism in mediating inverse risk for different cancers, studies of surrogate tissue telomere length and cancer risk have been contradictory and shown associations with short or long telomeres, or no effect[45–53]. TERT could also mediate risk through its telomere-independent functions that include transcriptional regulation and mitochondrial RNA polymerase activity (for review see Martinez *et al.*[54]). Other factors may contribute to the pleiotropic effects observed for rs36115365, including differential environmental exposures, regulatory effects through genes beyond *TERT*, interaction with additional risk variants and/or somatic mutations both within Region 2 and the larger *TERT/CLPTM1L* locus, or tissue-specific regulation of *ZNF148* and other transcription factors mediating *TERT* expression. Our findings represent the first steps in unravelling the complex functional consequences of carrying risk variants in Region 2 of chr5p15.33 and strongly indicate a major role for expression of *TERT* in influencing risk of multiple cancer types.

## Methods

**Studies.** Subjects were drawn from GWAS studies of four cancers: pancreatic cancer: PanScan I and II (3,525 cases and 3,642 control subjects; dbGaP Study Accession: phs000206.v5.p3) (refs 11,18); testicular germ cell tumours: NCI (581 cases and 1055 control subjects) and PENN (477 cases) (ref. 28) and PLCO controls (178 control subjects)[55]; lung cancer from the Transdisciplinary Research In Cancer of the Lung (TRICL) study with a total of 12,160 case and 16,838 control subjects from NCI (5,713/5,736), UK (1,952/5,200), IARC (2,533/3,791), MDACC (1,150/1,134), SLRI (331/499) and GERMANY (481/478) (ref. 56); and melanoma from the GenoMEL consortium with a total of 5,374 melanoma cases and 7,691 control subjects[29]. All participants provided informed written consent and all studies were reviewed and approved by institutional ethics review committees at the involved institutions. Participation of subjects in the PanScan GWAS was also reviewed by the NCI Special Studies Institutional Review Board. Each participating study obtained approval from its institutional review board (IRB) permitting data sharing in accordance with the NIH policy for Sharing of Data obtained in NIH-Supported or NIH-Conducted Genome Wide Association Studies. Analysis of melanoma GWAS was reviewed by The Northern and Yorkshire Research Ethics

Committee; each participating study obtained informed consent from study participants, approval from its local IRB as previously described[29]. Meta-analysis of data conducted for the Transdisciplinary Research in Cancer of the Lung has been approved as protocol numbers STUDY00023900 and STUDY00023602 which were approved by the Committee for the Protection of Human Subjects under the auspices of the Trustees of Dartmouth College Dartmouth-Hitchcock Medical Center. All studies were reviewed and approved by institutional ethics review committees at the involved institutions. Analysis of the testicular germ cell tumour GWAS was reviewed by the NCI Special Studies Institutional Review Board and the University of Pennsylvania IRB #4.

**Fine-mapping.** Imputation across 2 Mb of chr5p15.33 (250,000 to 2,250,000 bps, hg19) was performed using phased haplotypes from the 1000G reference set (Phase 1 integrated release 3, March 2012) and IMPUTE2 for pancreatic cancer[11,57] and testicular germ cell tumours[28]. Imputed SNPs with low MAF (<0.01) or low-quality scores (IMPUTE2 information score <0.5) were removed before the association analysis. Association analysis between SNPs and case control status were performed using the score test of the log additive genetic effect with covariate adjustment using SNPTEST as previously described[18]. Imputation and association analysis for melanoma was performed using 1000G (Phase 1 integrated release 3, March 2012) as previously described[29]. Imputation for lung cancer[7,56] was performed by using 1000G (Phase 1 integrated release 3, March 2012) with the same quality thresholds as described, followed by association analysis and conditional analysis using summary statistics from a meta-analysis of the six studies of TRICL with GCTA[58].

Overall, Region 2 was well-imputed. Within the pancreatic cancer GWAS data, all common 1000G variants ($n = 195$, MAF$\geq 0.01$) in Region 2 (defined as the genomic region between the two recombination hotspots at 1,306,281–1,367,281 in NCBI build Hg19) had imputation accuracy (INFO) scores above 0.3 (the lowest quality score was 0.48). The imputation quality for the set of nine Region 2 variants most significantly associated with pancreatic cancer risk was high in the PanScan GWAS studies, with quality scores (INFO) ranging from 0.82 to 0.96 (average 0.92). Similar imputation quality scores were observed for these SNPs in the lung cancer, TGCT, and melanoma GWAS (INFO range 0.82 to 0.98; average 0.94). In addition, imputation quality was high for all SNPs that were statistically correlated with rs36115365 in 1000 Genomes CEU data ($r^2 > 0.2$). In PanScan, only a single such 1,000 Genomes variant had an imputation quality score (INFO) below 0.8 (rs186156459; INFO = 0.79), suggesting that poor imputation quality did not lead to the exclusion of additional strong functional candidates from consideration. Similar imputation quality was likewise observed for the other cancer GWAS.

For completeness we assessed the newer 1000G (Phase 3, October 2014) reference dataset and noted an insertion/deletion variant (rs3030832) that was highly correlated to rs36115365 ($r^2 = 0.87$ in EUR). We therefore re-imputed the pancreatic cancer GWAS dataset[11,57] with the newer 1000G reference set to re-assess the association signal across Region 2 (defined as the genomic region between the two recombination hotspots at 1,306,281–1,367,281 in NCBI build Hg19) including this variant. rs36115365 became non-significant when analysis was conditioned on rs3030832, as was rs3030832, when analysis was conditioned on rs36115365 (Supplementary Table 2), indicating that this variant is among the highly correlated variants representing Region 2 and thus represents an additional strong functional candidate. We also observed seven additional variants with similar or slightly higher ORs as compared to rs36115365 (OR$_{MAX}$ = 1.42). To formally test if these seven variants represented potential functional variants in Region 2 we performed a series of conditional analyses. After the analysis was conditioned on rs36115365 we noted a large drop in significance for these seven variants while conditional analysis for each of the seven variants did not dramatically influence the significance or rs36115365 (Supplementary Table 2).

**Cell lines.** The human pancreatic cancer cell lines PANC-1 and MIA PaCa-2, and lung cancer cell lines A549 and NCI-H460 (purchased from ATCC) were maintained in Dulbecco's modified Eagle's medium (DMEM, Mediatech Inc, Herndon, VA) or RPMI 1640 (Mediatech Inc) supplemented with 10% fetal bovine serum (FBS, Life Technologies) or 10% FBS and 2.5% horse serum. The pancreatic cancer cell lines Panc 05.04, IMIM-PC-1, COLO 357 and IMIM-PC2 were a generous gift from Dr. Udo Rudloff, NCI, NIH, Bethesda, MD and Dr Francisco X. Real, The Spanish National Cancer Institute (CNIO) in Madrid, Spain. They were grown in RPMI 1640 supplemented with 15% FBS and Insulin (20U/ml) (Panc 05.04), RPMI 1640 supplemented with 10% FBS (COLO 357) or DMEM supplemented with 10% FBS (IMIM-PC-1, IMIM-PC2). The testicular germ cell tumour cell lines NTERA-2 [NT2/D1] and 2102Ep (generously donated by Dr. Roelof Koster, NCI, NIH) were maintained in RPMI 1640 supplemented with 10% FBS. The two melanoma cell lines, UACC1113 and UACC903, were obtained from the University of Arizona Cancer Center and grown in RPMI 1640 supplemented with 10% FBS and 25 mM HEPES. Cell lines were tested for authentication (October 2014) with a panel of short tandem repeats (STR) using the Identifiler kit (Life Technologies) and compared with the ATCC and the DSMZ (German Collection of Microorganisms and Cell Cultures) STR Profile Databases. All cell lines with profiles in either database matched (UACC1113, UACC903, 2102Ep, IMIM-PC-1 and IMIM-PC2 did not have profiles listed). The 2012Ep profile matched a previously published profile[59]. The cells were routinely tested for Mycoplasma and

were negative on each occasion. None of the cell lines used are on the NCI or ICLAC lists of misidentified cells.

**RNA and genomic DNA isolation.** RNA was extracted using an RNeasy Plus Mini Kit (Qiagen). Quality and quantity of RNA was assessed in an Agilent 2100 Bioanalyzer (Agilent Technologies); only samples with RIN scores > 9.0 were used. Genomic DNA was isolated using the ZR genomic DNA (D3050, ZYMO Research) and assessed by Nanodrop 8000 (Thermo Scientific).

**EMSAs and ChIP.** Nuclear extracts were purchased from Active Motif (PANC-1, MIA PaCa-2) or alternatively generated using a Nuclear Extraction Kit (A549, NCI-H460, UACC1113, UACC903, NTERA-2 and 2102Ep) (10009277, Cayman) according to the manufacturer's instructions. Recombinant human ZNF148 protein was purchased from Origene (TP602963, Origene). Oligos (30-36 nt, Invitrogen, listed in Supplementary Table 3) were labelled with a biotin 3′end-labelling kit (NEB). Forward and reverse oligos were then annealed to create double stranded 3′-end labelled or unlabelled probes. EMSAs were performed (Pierce) by incubating labelled probes for 20 min on ice with nuclear extracts (10 μg per reaction). Competition experiments were performed by adding 10–100 fold more unlabelled competitor than labelled probes. Supershift experiments were carried out by mixing 1 and 2 μg anti-ZNF148/ZBP-89 (sc-48811X, 200 μg per 0.1 ml, Santa Cruz), anti-VEZF1/ZNF161 (sc-98278X, 200 μg per 0.1 ml, Santa Cruz), anti-ZNF281 (sc-166933X, 200 μg per 0.1 ml, Santa Cruz), anti-ZNF740 (sc-324747, 200 μg ml$^{-1}$, Santa Cruz) or anti-IgG (sc-52001, 100 μg ml$^{-1}$, Santa Cruz) antibodies with nuclear extracts for 20 min at room temperature before adding labelled probe. Recombinant human ZNF148 protein was added at 90, 360 and 630 ng to the reaction, separately. The resulting protein complexes were resolved on 4–20% TBE gels (Bio-Rad), transferred to Biodyne B membranes (VWR), crosslinked (Stratagene UV Stratalinker 1800), and detected using streptavidin-HRP conjugate and a chemiluminescent substrate (20148 LightShift Kit; Pierce).

Chromatin-immunoprecipitation (ChIP) was performed with the ChIP-IT high Sensitivity kit (Active Motif) according to the manufacturer's protocol using cells ($\sim 2 \times 10^7$) from each cell line of the following genotypes at rs36115365: CC (COLO 357 and IMIM-PC2, CG (Panc 05.04 and A549) and GG (Mia PaCa-2). An anti-ZNF148/ZBP-89 antibody (4 μg, sc-48811X, 200 μg per 0.1 ml, Santa Cruz) or nonspecific IgG (4 μg, sc-2027X, 200 μg per 0.1 ml, Santa Cruz) were used for ChIP on 12–24 μg chromatin from each cell line. Purified pulled-down DNA was assayed by nine SYBR Green qPCR amplicons for enrichment of target sites using primers listed in Supplementary Table 4. A TaqMan genotyping assay for rs36115365 (C_470504_10, Life Technologies) was used to quantify the C and G alleles in immunoprecipitated DNA samples in seven independent experiments (Supplementary Fig. 12g). A paired two sided T-test was applied to C- and G-allele signals (normalized to input DNA) in order to assess significance of enrichment of the C versus G allele at rs36115365. The specificity of the ZNF148 antibody was tested by western blot analysis with and without siRNA mediated knockdown of ZNF148. GAPDH (ab37168, 1 mg ml$^{-1}$, Abcam) was used as a loading control (Supplementary Fig. 18).

**Proteome-wide analysis of disease-associated SNPs.** Nuclear extract collection and DNA pulldowns were performed essentially as described previously for both PANC-1 and UACC903 cell lines, using biotin-tagged oligo probes consisting of 20 bp on either side of rs36115365 (refs 60,61). After PBS washes, beads were resuspended in 50 μl 100 mM TEAB buffer, reduced, alkylated and digested with trypsin overnight. Then, digested peptides were labelled using dimethyl chemical labelling as described previously[62,63]. Experiments were performed in duplicate using label-swapping, and separately conducted using poly-dAdT competitor only, as well as using both poly-dAdT and poly-dIdC competitor. Data analysis was performed using MaxQuant (version 1.3.0.5) as described previously, using dithiomethane instead of carbamidomethylation as a fixed modification[32,64].

**Luciferase cloning and expression analysis.** The genomic region containing and surrounding rs36115365 (240 bps) was PCR-amplified (primers listed in Supplementary Table 5) from HapMap CEU DNA samples with the appropriate genotypes to obtain clones with each genotype, and cloned into the NheI and BglII sites of the pGL4.23[luc2/minP] (Promega) luciferase vector in the 5′-to-3′ or 3′-to-5′ orientation. Plasmid inserts were sequence-verified to contain the correct inserts and genotypes. The forward (F) orientation of the inset is the same as the genomic orientation. The Firefly reporter plasmids (and a Renilla luciferase control vector) were co-transfected into pancreatic (PANC-1, MIA PaCa-2), melanoma (UACC903, UACC1113), lung (A549, NCI-H460) and testicular (NTERA-2, 2102Ep) cancer cell lines at ∼70% confluence using Lipofectamine 2000 (Life Technologies). Luciferase activity was measured 36 h after transfection with the Dual Luciferase Reporter Assay System (Promega). Firefly luciferase activity was normalized to Renilla luciferase activity, and graphed as compared to the empty luciferase vector. Experiments were performed in triplicate and repeated at least three times. A T-test was used to assess significance for differences in luciferase activity.

**Region targeted siRNAs to rs36115365 regulatory locus.** On-target antisense enhanced siRNAs targeting the locus encompassing rs36115365 were designed by using an siRNA design tool (http://dharmacon.gelifesciences.com/) and ordered from Dharmacon RNAi and Gene Expression in GE Healthcare and listed in Supplementary Table 6. No siRNAs were designable to directly overlap with rs36115365; the location of the nearest siRNA (siRNA3) was 8 bp from this variant. The siRNAs were introduced to cell lines by using RNAiMAX (Life Technologies) at a final concentration of 15 nM. RNA was extracted 48 h after transfection and reverse transcribed to cDNA by the SuperScript III First-Strand Synthesis System for RT-PCR (Life Technologies). Expression of target genes was determined on the cDNA by RT-qPCR TaqMan gene expression assays as described below. Experiments were performed in triplicate and repeated at least three times. We first tested 8 siRNAs in 4 cell lines. Three of the 8 siRNAs inhibited TERT expression in all four cell lines (PANC-1, A549, NTERA-2 and UACC903) whereas none of the 8 siRNAs inhibited CLPTM1L, ACTB or GAPDH expression. Thus, the inhibition of TERT expression by 3 out of 8 siRNAs versus 0 out of 8 for the other three genes gives rise to a Fisher's Exact test P value of 0.011, indicating that the inhibition of TERT is specific.

**siRNA-mediated knockdown of ZNF148 and TERT mRNA.** ON-TARGETplus Human SMARTpool siRNAs to ZNF148 (cat# L-012658-00-0005), VEZF1 (ZNF161; cat# L-019623-00-0005), ZNF281 (cat# L-006958-00-0005), ZNF740 (cat# L-030075-02-0005), and TERT (cat# L-003547-00-0005) were purchased from Dharmacon RNAi and Gene Expression in GE Healthcare. To assess possible off-target effects for the ZNF148 siRNAs we also purchased each of the four siRNAs from the SMARTpool separately and tested their effects on ZNF148 and TERT expression. All four siRNAs inhibited both ZNF148 and TERT expression indicating that off-target effects are not likely to explain our findings (Supplementary Fig. 14). Transfection, RNA purification, cDNA generation and expression analysis procedures were as described above for the region-targeted siRNA assay, except that RNA was isolated 72 h after transfection. Experiments were performed in triplicate and repeated at least three times.

**Real-time quantitative PCR.** Gene expression levels were quantified by quantitative real-time PCR using TaqMan assays for TERT (Hs00972656_m1), CLPTM1L (Hs00363947_m1), ACTB (cat# 4333762), ZNF148 (Hs01070570_m1), and GAPDH (cat# 4333764) from Life Technologies. Gene expression levels of TERT, CLPTM1L and ACTB were normalized to GAPDH, while expression of GAPDH was normalized to ACTB. Allele-specific TERT expression was determined using an allelic discrimination TaqMan assay for rs2736098 (assay C_26414916_20, Life Technologies), and the gene expression of each allele of TERT was also normalized to the gene expression of GAPDH. Each experiment was performed in triplicate and repeated three times. Significance was assessed using a Student's two-tailed T-test (labelled significant if P < 0.01).

**Telomerase activity and telomere length.** The telomeric repeat amplification protocol (TRAP) (ref 20) was used to evaluate telomerase activity according to the manufacturer's guidelines (Millipore, #S7700). siRNAs targeting the gene regulatory region (siRNA3), ZNF148 (cat# L-012658-00-0005), TERT (cat# L-003547-00-0005) and a scrambled control siRNA (sequence listed in Supplementary Table 6) were administered to MIA PaCa-2, A549, UACC903 and NTERA-2 cells at a final concentration of 15 nM for 72 h. At that time the cells were harvested and whole cell extract prepared using CHAPS (3-[(3-cholamidopropyl)-dimethylammonio]-1-propanesulfonate) solution. The Bradford assay kit (Bio-Rad) was used to determine total protein concentration. Equal amounts of protein extracts were used to add telomeric repeats (GGTTAG) onto 3′ end of substrate oligosnucleotide (TS) at 37 °C for 30 min followed by 30 cycles of TRAP PCR and separation of PCR products on 12.5% non-denatured PAGE gels. The gels were stained with SYBR Gold Nucleic Acid Gel Stain (Life Technologies, #S-11494).

For rescue experiments, we attempted to create cell lines devoid of ZNF148 using CRISPR. Extensive screening of clones revealed none with homozygous loss of ZNF148, consistent with an essential role for ZNF148, further supported by the observed embryonic lethality of homozygous ZNF148 knock-out mice (International Mouse Phenotyping Consortium, IMPC Data Coordination Centre, MRC Harwell Institute, Biocomputing, Harwell Campus, https://www.mousephenotype.org/data/charts?accession=MGI:1332324&allele_accession_id=MGI:5636955 &zygosity=homozygote ¶meter_stable_id= IMPC_VIA_001_001&pipeline_stable_id=BCM_001). Rescue experiments were instead performed by depletion of endogenous ZNF148 expression using an siRNA targeting the 3′-UTR of ZNF148 (designed using the 3′-UTR sequences of ZNF148; sense: AUGGAGAACUUGAUGCAAU; antisense: AUUGCAUCAAGUUCUC CAU) and reintroduction of exogenous ZNF148 expression. Human ZNF148 ORF (ORigene TrueORF RC222687) was cloned into the pDest-663 (derived from pFUGW) lentiviral expression vector and sequence verified. For lentivirus production, lentiviral vectors were co-transfected into HEK293FT cells with packaging vectors psPAX2, pMD2-G and pCAG4-RTR2. Virus was collected two days after transfection and concentrated by Vivaspin, before infecting MIA PaCa-2

and A549 cells. Seventy-two hours after delivery of the siRNA, *ZNF148* and *TERT* expression, and telomerase activity, were assessed as described above.

To assay effects on telomere length, siRNA3, *ZNF148* siRNA (cat# L-012658-00-0005), *TERT* siRNA (cat# L-003547-00-0005) and a scrambled siRNA (Supplementary Table 6) were administered to MIA PaCa-2 and A549 cells at a final concentration of 15 nM. The cells were re-transfected with siRNAs every four days and genomic DNA extracted 20 days later using the DNeasy Blood and Tissue Kit (Qiagen, #69506). Telomere length was then determined by qPCR by comparing telomere repeat sequence copy number to a single-copy gene (RPLP0) copy number in a given sample using telomere repeat-specific primers as previously described[65]. The assays were performed in triplicate and repeated three times.

**Data availability.** The authors declare that all data supporting the findings of this study are available within the article and its supplementary information files or from the corresponding authors upon reasonable request. Pancreatic cancer GWAS data is available from dbGAP (phs000206.v5.p3). Testicular germ cell tumour fine-mapping data are available from the corresponding authors upon reasonable request. The lung cancer fine-mapping data that supports the findings of this study are available from the corresponding authors upon reasonable request and are currently being processed for availability through dbGAP. The melanoma fine-mapping data that support the findings of this study are available from MMI (M.M.Iles@leeds.ac.uk) on reasonable request subject to specific consent for contributing cohorts.

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

## Acknowledgements

We acknowledge the contribution of the staff of the Cancer Genomics Research Laboratory (CGR, NCI), as well as Mr Timothy Myers and Lea Jessop (LTG, DCEG, NCI) for their invaluable help throughout the project. We are grateful to Dr Francisco X. Real, The Spanish National Cancer Institute (CNIO) in Madrid, Spain for providing reagents. We are grateful to Dr Dom Esposito, Frederick National Laboratory for Cancer Research, Frederick, MD, USA, for cloning for lentiviral production. We thank Paul D.P. Pharoah, Department of Oncology, University of Cambridge, Cambridge, UK and Centre for Cancer Genetic Epidemiology, Department of Public Health and Primary Care, Strangeways Research Laboratory, Cambridge, UK for providing GWAS data. Furthermore, we thank Dr Stephen Chanock for his advice, encouragement and critical reading of the manuscript. This work was supported by the Intramural Research Program of the US National Institutes of Health (NIH), National Cancer Institute, by contract number HHSN261200800001E, by Cancer Research UK for SEARCH (C490/A10124). Funding for M.V. was provided by the Netherlands Organization for Scientific Research (NWO Gravitation Program Cancer Genomics Netherlands). Funding for M.M. was provided by a grant from the Marie Curie Initial Training Network (ITN) DevCom (FP7, grant number 607142). The content of this publication does not necessarily reflect the views or policies of the Department of Health and Human Services, nor does mention of trade names, commercial products, or organizations imply endorsement by the US Government.

## Author contributions

K.M.B. and L.T.A. conceived and directed the project, with contributions from M.V. and T.A. EMSAs, supershift experiments and reporter assays were performed by J.J., I.C., A.T., M.D., J.W.H., J.T. and Q.L. Transcriptional gene silencing, gene knockdown, telomerase activity and telomere length assays were performed by J.F. ChIP experiments were performed by J.J. and M.X. Quantitative mass-spectrometry experiments were led by M.V. and performed by M.M., P.W.T.C.J.; additional proteomics contributions were made by T.A., M.O. and S.D. Screening and characterization of cell lines used in this work was performed by J.J., I.C., J.F., R.K., J.C. and M.K. Fine-mapping and associated analyses of were performed by L.T.A., K.M.B., M.Z., Z.W., M.M.I., M.T.L., C.I.A., G.M.P. R.S.S., P.K., B.M.W., S.O., K.M., P.A.K., N.C., J.H.B., A.M.D., J.C.T., J.A.N.B., D.T.B., K.N., as well as the PanScan, TRICL and GenoMEL Consortia. K.M.B. and L.T.A. wrote the manuscript, with assistance from J.F. and J.J.

## Additional information

**Competing interests:** The authors declare no competing financial interests.

## PanScan Consortium

Federico Canzian[17], Charles Kooperberg[18], Zhaoming Wang[19,20], Alan A. Arslan[21,22,23], Paige M. Bracci[24], Julie Buring[25,26], Eric J. Duell[27], Steven Gallinger[28], Eric J. Jacobs[29], Aruna Kamineni[30], Stephen Van Den Eeden[30], Alison P. Klein[31,32], Laurence N. Kolonel[33], Donghui Li[34], Sara H. Olson[35], Harvey A. Risch[36], Howard D. Sesso[25,26,37], Kala Visvanathan[38], Wei Zheng[39,40], Demetrius Albanes[19], Melissa A. Austin[41], Marie-Christine Boutron-Ruault[42,43,44], H. Bas Bueno-de-Mesquita[45,46,47], Michelle Cotterchio[48,49], J. Michael Gaziano[25,26,50], Edward L. Giovannucci[37,51,52], Michael Goggins[53,54,55], Myron Gross[56], Manal Hassan[34], Kathy J. Helzlsouer[57], Elizabeth A. Holly[24], David J. Hunter[58,59,60], Mazda Jenab[61], Rudolf Kaaks[62], Timothy J. Key[63], Kay-Tee Khaw[64], Vittorio Krogh[65], Robert C. Kurtz[66], Andrea LaCroix[18], Loic Le Marchand[67], Satu Mannisto[68], Alpa V. Patel[29], Petra H.M. Peeters[69,70], Elio Riboli[47], Xiao-Ou Shu[39,40], Malin Sund[71], Mark Thornquist[18], Anne Tjønneland[72], Geoffrey S. Tobias[19], Dimitrios Trichopoulos[37,73,74], Jean Wactawski-Wende[75], Herbert Yu[67], Kai Yu[19], Anne Zeleniuch-Jacquotte[22,23], Robert Hoover[19], Patricia Hartge[19], Charles Fuchs[51,76] & Stephen J. Chanock[19,20]

[17]Genomic Epidemiology Group, German Cancer Research Center (DKFZ), Heidelberg, Germany; [18]Division of Public Health Sciences, Fred Hutchinson Cancer Research Center, Seattle, Washington, USA; [19]Division of Cancer Epidemiology and Genetics, National Cancer Institute, National Institutes of Health,

Bethesda, Maryland, USA; [20]Cancer Genomics Research Laboratory, National Cancer Institute, Division of Cancer Epidemiology and Genetics, Leidos Biomedical Research, Inc., Frederick National Laboratory for Cancer Research, Frederick, Maryland, USA; [21]Department of Obstetrics and Gynecology, New York University School of Medicine, New York City, New York, USA; [22]Department of Environmental Medicine, New York University School of Medicine, New York City, New York, USA; [23]New York University Cancer Institute, New York City, New York, USA; [24]Department of Epidemiology and Biostatistics, University of California San Francisco, San Francisco, California, USA; [25]Division of Preventive Medicine, Department of Medicine, Brigham and Women's Hospital and Harvard Medical School, Boston, Massachusetts, USA; [26]Division of Aging, Department of Medicine, Brigham and Women's Hospital and Harvard Medical School, Boston, Massachusetts, USA; [27]Unit of Nutrition, Environment and Cancer, Cancer Epidemiology Research Program, Bellvitge Biomedical Research Institute (IDIBELL), Catalan Institute of Oncology (ICO), Barcelona, Spain; [28]Samuel Lunenfeld Research Institute, Mount Sinai Hospital, Toronto, Ontario, Canada; [29]Epidemiology Research Program, American Cancer Society, Atlanta, Georgia, USA; [30]Group Health Research Institute, Seattle, Washington, USA; [31]Department of Oncology, the Johns Hopkins University School of Medicine, Baltimore, Maryland, USA; [32]Department of Epidemiology, the Bloomberg School of Public Health, Baltimore, Maryland, USA; [33]The Cancer Research Center of Hawaii (retired), Honolulu, Hawaii, USA; [34]Department of Gastrointestinal Medical Oncology, University of Texas M.D. Anderson Cancer Center, Houston, Texas, USA; [35]Department of Epidemiology and Biostatistics, Memorial Sloan-Kettering Cancer Center, New York City, New York, USA; [36]Department of Chronic Disease Epidemiology, Yale School of Public Health, New Haven, Connecticut, USA; [37]Department of Epidemiology, Harvard School of Public Health, Boston, Massachusetts, USA; [38]Johns Hopkins Bloomberg School of Public Health, Baltimore, Maryland, USA; [39]Department of Medicine, Vanderbilt University Medical Center, Nashville, Tennessee, USA; [40]Vanderbilt-Ingram Cancer Center, Vanderbilt University Medical Center, Nashville, Tennessee, USA; [41]Department of Epidemiology, School of Public Health, University of Washington, Seattle, Washington, USA; [42]Inserm, Centre for Research in Epidemiology and Population Health (CESP), U1018, Nutrition, Hormones and Women's Health Team, Villejuif F-94805, France; [43]University Paris Sud, UMRS 1018, Villejuif F-94805, France; [44]IGR, Villejuif F-94805, France; [45]National Institute for Public Health and the Environment (RIVM), Bilthoven, The Netherlands; [46]Department of Gastroenterology and Hepatology, University Medical Centre Utrecht, Utrecht, The Netherlands; [47]Department of Epidemiology and Biostatistics, School of Public Health, Imperial College London, London, UK; [48]Cancer Care Ontario, University of Toronto, Toronto, Ontario, Canada; [49]Dalla Lana School of Public Health, University of Toronto, Toronto, Ontario, Canada; [50]Massachusetts Veteran's Epidemiology, Research, and Information Center, Geriatric Research Education and Clinical Center, Veterans Affairs Boston Healthcare System, Boston, Massachusetts, USA; [51]Channing Division of Network Medicine, Department of Medicine, Brigham and Women's Hospital, and Harvard Medical School, Boston, Massachusetts, USA; [52]Department of Nutrition, Harvard School of Public Health, Boston, Massachusetts, USA; [53]Department of Pathology, Sidney Kimmel Cancer Center and Johns Hopkins University, Baltimore, Maryland, USA; [54]Department of Medicine, Sidney Kimmel Cancer Center and Johns Hopkins University, Baltimore, Maryland, USA; [55]Department of Oncology, Sidney Kimmel Cancer Center and Johns Hopkins University, Baltimore, Maryland, USA; [56]Laboratory of Medicine and Pathology, University of Minnesota, Minneapolis, Minnesota, USA; [57]Prevention and Research Center, Mercy Medical Center, Baltimore, Maryland, USA; [58]Department of Medicine, Brigham and Women's Hospital and Harvard Medical School, Boston, Massachusetts, USA; [59]Harvard School of Public Health, Boston, Massachusetts, USA; [60]Harvard Medical School, Boston, Massachusetts, USA; [61]International Agency for Research on Cancer, Lyon, France; [62]Division of Cancer Epidemiology, German Cancer Research Center (DKFZ), Heidelberg, Germany; [63]Cancer Epidemiology Unit, University of Oxford, Oxford, UK; [64]School of Clinical Medicine, University of Cambridge, UK; [65]Epidemiology and Prevention Unit, Fondazione IRCCS Istituto Nazionale dei Tumori, Milan, Italy; [66]Department of Medicine, Memorial Sloan-Kettering Cancer Center, New York City, New York, USA; [67]Cancer Epidemiology Program, University of Hawaii Cancer Center, Honolulu, Hawaii, USA; [68]National Institute for Health and Welfare, Department of Chronic Disease Prevention, Helsinki, Finland; [69]Julius Center for Health Sciences and Primary Care, University Medical Center Utrecht, Utrecht, The Netherlands; [70]Department of Epidemiology and Biostatistics, School of Public Health, Imperial College London, London, UK; [71]Department of Surgical and Peroperative Sciences, Umeå University, Umeå, Sweden; [72]Institute of Cancer Epidemiology, Danish Cancer Society, Copenhagen, Denmark; [73]Bureau of Epidemiologic Research, Academy of Athens, Athens, Greece; [74]Hellenic Health Foundation, Athens, Greece; [75]Department of Social and Preventive Medicine, University at Buffalo, Buffalo, New York, USA; [76]Department of Medical Oncology, Dana-Farber Cancer Institute, Boston, Massachusetts, USA.

## TRICL Consortium

Victoria Stevens[77], Demetrios Albanes[78], Neil E. Caporaso[78], Paul Brennan[79], James McKay[79], Xifeng Wu[80], Rayjean J. Hung[81], John R. McLaughlin[82], Heike Bickeboller[83], Angela Risch[84,85,86], Erich Wichmann[87] & Richard S. Houlston[88]

[77]American Cancer Society, Inc., Atlanta, Georgia 30303, USA; [78]Division of Cancer Epidemiology and Genetics, National Cancer Institute, National Institutes of Health, Bethesda, Maryland 20892, USA; [79]International Agency for Research on Cancer, World Health Organization, 69372, Lyon CEDEX 08, France; [80]Department of Epidemiology, The University of Texas MD Anderson Cancer Center, Houston, Texas 77230, USA; [81]Lunenfeld-Tanenbaum Research Institute of Mount Sinai Hospital, University of Toronto, Toronto, Canada M5G 1X5; [82]University of Toronto, Dalla Lana School of Public Health, Toronto, Ontario, Canada M5S 1A1; [83]Department of Genetic Epidemiology, University Medical Center, Georg-August-University, Göttingen 37073, Germany; [84]Division of Epigenomics and Cancer Risk Factors, DKFZ-German Cancer Research Center, Heidelberg 69121, Germany; [85]Division of Cancer Epidemiology, DKFZ-German Cancer Research Center, Heidelberg 69121, Germany.; [86]Division of Cancer Research and Epigenetics, Department of Molecular Biology, University of Salzburg, Salzburg 5020, Austria; [87]Helmholtz Zentrum München, Institut für Epidemiologie I, Neuherberg D-85764, Germany; [88]Division of Genetics and Epidemiology, The Institute of Cancer Research, London, UK; Division of Molecular Pathology, The Institute of Cancer Research, Surrey SM2 5NG, UK.

## GenoMEL Consortium

Graham Mann[89,90], John Hopper[91], Joanne Aitken[92], Bruce Armstrong[93], Graham Giles[94], Elizabeth Holland[90], Richard Kefford[89,95], Anne Cust[89,93], Mark Jenkins[91], Helen Schmid[90], Susana Puig[96], Paula Aguilera[97], Celia Badenas[98], Alicia Barreiro[96], Cristina Carrera[96], Daniel Gabriel[96], Pol Gimenez Xavier[96], Pablo Iglesias-Garcia[96], Josep Malvehy[96], Montse Mila[98], Ramon Pigem[99], Miriam Potrony[96], Joan-Anton Puig Butille[98], Gemma Tell-Marti[96], Nicholas K. Hayward[100], Nicholas G. Martin[100],

Grant Montgomery[101], David L. Duffy[100], David C. Whiteman[100], Stuart MacGregor[100], Donato Calista[102], Giorgio Landi[102], Paola Minghetti[102], Fabio Arcangeli[102], Pier Alberto Bertazzi[103], Paola Ghiorzo[104], Giovanna Bianchi-Scarra[104], Lorenze Pastorino[104], William Bruno[104], Virginia Andreotti[104], Paola Queirolo[105], Francesco Spagnolo[105], Rona MacKie[106], Julie Lang[107], Nelleke Gruis[108], Frans A. van Nieuwpoort[108], Coby Out[108], Wilma Bergman[108], Nicole Kukutsch[108], Jan Nico Bouwes Bavinck[108], Bert Bakker[109], Nienke van der Stoep[109], Jeanet ter Huurne[109], Han van der Rhee[110], Marcel Bekkenk[111], Dyon Snels[112], Marinus van Praag[112], Lieve Brochez[113], Rianne Gerritsen[114], Marianne Crijns[114], Hans Vasen[115], Bart Janssen[115], Christian Ingvar[116], Håkan Olsson[116], Göran Jönsson[116], Åke Borg[116], Katja Harbst[116], Kari Nielsen[117], Anita Schmidt Zander[116], Anders Molven[118,119], Per Helsing[120], Per Arne Andresen[121], Helge Rootwelt[122], Lars A. Akslen[119,123], Brigitte Bressac-de Paillerets[124], Florence Demenais[125], Marie-Francoise Avril[126,127], Valerie Chaudru[128], Patricia Jeannin[125], Fabienne Lesueur[129], Eve Maubec[125], Hamida Mohamdi[125], Myriam Bossard[125], Amaury Vaysse[125], Francoise Boitier[130], Olivier Caron[131], Frederic Caux[132], Stephane Dalle[133,134], Olivier Dereure[135], Dominique Leroux[136], Ludovic Martin[137], Christine Mateus[138], Caroline Robert[139], Dominique Stoppa-Lyonnet[140], Luc Thomas[133,134], Eva Wierzbicka[141,142], David E. Elder[143], Michael Ming[143], Nandita Mitra[144], Tadeusz Debniak[145], Jan Lubinski[145], Marko Hocevar[146], Srdjan Novakovic[147], Barbara Peric[146], Petra Skerl[147], Johan Hansson[148], Veronica Höiom[148], Eitan Friedman[149,150], Esther Azizi[151], Orna Baron-Epel[152], Alon Scope[151,153], Felix Pavlotsky[153], Irit Cohen-Manheim[154], Yael Laitman[155], Mark Harland[156], Juliette Randerson-Moor[156], Jon Laye[156], John Davies[156], Jeremie Nsengimana[156], Sally O'Shea[156], May Chan[156], Jo Gascoyne[156], Margaret A. Tucker[157], Alisa M. Goldstein[157] & Xiaohong R. Yang[157]

[89]Melanoma Institute Australia, Wollstonecraft, New South Wales 2065, Australia; [90]Centre for Cancer Research, Westmead Millennium Institute for Medical Research, Westmead, New South Wales 2145, Australia; [91]Centre for Epidemiology and Biostatistics, Melbourne School of Population and Global Health, The University of Melbourne, Victoria 3010, Australia; [92]Viertel Centre for Research in Cancer Control, Cancer Council Queensland, Spring Hill, Queensland 4004, Australia; [93]Cancer Epidemiology and Prevention Research, Sydney School of Public Health, The University of Sydney, Sydney, New South Wales 2006, Australia; [94]Cancer Epidemiology Centre, Cancer Council Victoria, Melbourne 3004, Australia; [95]Macquarie University Health Sciences Centre, Macquarie University, Sydney, New South Wales 2109, Australia; [96]Melanoma Unit, Department of Dermatology Hospital Clínic de Barcelona, IDIBAPS, Villarroel 170, Barcelona 08036, Spain; [97]Photodermatology Department, Hospital Clinic de Barcelona, Universitat de Barcelona, Barcelona 08036, Spain; [98]Biochemistry and Molecular Genetics Department, Hospital Clinic de Barcelona, Barcelona 08036, Spain; [99]Melanoma Unit, Department of Dermatology Hospital Clínic de Barcelona, IDIBAPS, Villarroel 170, Barcelona 08036, Spain; [100]Department of Genetics & Computational Biology, QIMR Berghofer Medical Research Institute, Herston, Queensland 4006, Australia; [101]The Institute for Molecular Bioscience, University of Queensland, St Lucia, Queensland 4072, Australia; [102]Department of Dermatology, Maurizio Bufalini Hospital, Viale Ghirotti, 286, Cesena FC, Italy; [103]Department of Occupational and Environmental Health, EPOCA, Research Center for Occupational, Clinical and Environmental Epidemiology, University of Milan, Milan 20122, Italy; [104]Genetics of Rare Cancers Unit, Department of Internal Medicine and Medical Specialties, University of Genoa and IRCCS AOU San Martino-IST Istituto Nazionale per la Ricerca sul Cancro, Genoa 16132, Italy; [105]Medical Oncology Unit, IRCCS AOU San Martino-IST Istituto Nazionale per la Ricerca sul Cancro, Genoa 16132, Italy; [106]Department of Public Health, Glasgow University, Glasgow G12, UK; [107]Research & Development Central Office, West Glasgow Ambulatory Care Hospital, Glasgow G3 8SJ, UK; [108]Department of Dermatology, Leiden University Medical Center, 2333 za Leiden, The Netherlands; [109]Department of Clinical Genetics, Laboratory for Diagnostic Genome Analysis Clinical Genetics (LDGA), Leiden University Medical Center, 2333 ZC Leiden, The Netherlands; [110]Haga ziekenhuis, Department of Dermatology, 2504 LN The Hague, The Netherlands; [111]Department of Dermatology, Academic Medical Center, 1105 AZ Amsterdam, The Netherlands; [112]Department of Dermatology, Franciscus Gasthuis, Kleiweg 500, Rotterdam 3045PM, The Netherlands; [113]Department of Dermatology, Ghent University Hospital, Belgium and Cancer Research Institute Ghent (CRIG), Ghent 9000, Belgium; [114]Department of Dermatology, Radboud University Medical Center, Nijmegen 500 HB, The Netherlands; [115]Netherlands Foundation for the Detection of Hereditary Tumors, Leiden 2333AA, The Netherlands; [116]Department of Oncology, Clinical Sciences, Lund University, Lund 221 85, Sweden; [117]Department of Dermatology Helsingborg, Clinical Sciences, Lund University, Lund 221 85, Sweden; [118]Gade Laboratory for Pathology, Department of Clinical Medicine, University of Bergen, Bergen N-5020, Norway; [119]Department of Pathology, Haukeland University Hospital, Bergen 5021, Norway; [120]Department of Dermatology, Oslo University Hospital, Oslo NO-0424, Norway; [121]Department of Pathology, Oslo University Hospital, Oslo NO-0424, Norway; [122]Department of Medical Biochemistry, Oslo University Hospital, Oslo NO-0424, Norway; [123]Center for Cancer Biomarkers, Department of Clinical Medicine, University of Bergen, Bergen N-5020, Norway; [124]Gustave Roussy, Université Paris-Saclay, Département de Biologie et Pathologie Médicales, Villejuif F-94805, France; [125]Genetic Variation and Human Diseases Unit, UMR-946, INSERM, Université Paris Diderot, Université Sorbonne Paris Cité, Paris F-75007, France; [126]Hôpital Cochin, Service de Dermatologie, Assistance Publique-Hôpitaux de Paris, Paris 75014, France; [127]Faculté de Médecine, Université Paris Descartes, 12 Rue de l'École de Médecine, Paris 75006, France; [128]GenHotel-EA3886, Evry University, Evry, France; [129]Inserm U900, Institut Curie, Mines ParisTech, PSL University, 26 rue d'Ulm, Paris F-75248, France; [130]Dermatologue, Hopital Cochin, Paris 75006, France; [131]Gustave Roussy, Université Paris-Saclay, Département de Médecine Oncologique, Villejuif F-94805, France; [132]Department of Dermatology, Hopital Avicenne, Bibigny 93000, France; [133]Cancer Research Center of Lyon, INSERM U1052, CNRS UMR5286, Department of Dermatology, HCL-Cancer Institute, Lyon 69008, France; [134]Claude Bernard Lyon 1 University Centre Hospitalier Lyon Sud, Pierre Bénite 69495, France; [135]Department of Dermatology, HCL-Cancer Institute, Lyon 69008, France; [136]Institut Curie, Department of Tumour Biology, INSERM U830, Paris, France; [137]Department of Dermatology, Angers University Hospital, Angers 49933, France; [138]Dermatology Unit, Oncology department, Gustave Roussy, Campus Cancer Grand Paris, Villejuif 94805, France; [139]Dermatology in Paris

Sud University, Gustave Roussy, Villejuif 94801, France; [140]Département de Biologie des tumeurs–Service Génétique, Institut Curie, Paris 75248, France; [141]Dermatology Department, CHU de Poitiers, Poitiers 86021, France; [142]Laboratoire Inflammation, Tissus Epithéliaux et Cytokines, EA 4331 Université de Poitiers, Pôle Biologie Santé, Poitiers 86073, France; [143]Pathology & Laboratory Medicine, Anatomic Pathology Division, University of Pennsylvania Perelman School of Medicine, Philadelphia, Pennsylvania 19104, USA; [144]Department of Biostatistics and Epidemiology, University of Pennsylvania Perelman School of Medicine, Philadelphia, Pennsylvania 19104, USA; [145]Depatment of Genetics and Patomorphology, Pomeranian Medical University, Szczecin 70-115, Poland; [146]Department of Surgical Oncology, Institute of Oncology, Ljubljana 1000, Slovenia; [147]Department of Molecular Diagnostics, Institute of Oncology, Ljubljana 1000, Slovenia; [148]Department of Oncology-Pathology, Karolinska institutet, Karolinska University Hospital, Stockholm 171 76, Sweden; [149]The Susanne Levy Gertner Oncogenetics Unit, The Danek Gertner Institute of Human Genetics, Chaim Sheba Medical Center, Tel-Hashomer 52621, Israel; [150]Department of Internal Medicine and Department of Human Genetics and Biochemistry, Sackler School of Medicine, Tel Aviv University, Tel Aviv, Ramat Aviv 69978, Israel; [151]Department of Dermatology, Sackler Faculty of Medicine, Tel Aviv University, Tel Aviv, Ramat Aviv 69978, Israel; [152]School of Public Health, Faculty of Welfare and Health Studies, Haifa University, Mount Carmel, Haifa 31905, Israel; [153]Department of Dermatology, Chaim Sheba Medical Center- Tel Hashomer, Ramat Gan 52621, Israel; [154]Hebrew University-Hadassah Braun School of Public Health and Community Medicine, Ein Kerem, Jerusalem 91120, Israel; [155]The Susanne Levy Gertner Oncogenetics Unit, Institute of Human Genetics, Chaim Sheba Medical Center- Tel Hashomer, Ramat Gan 52621, Israel; [156]Section of Epidemiology and Biostatistics, Leeds Institute of Cancer and Pathology, University of Leeds, Leeds LS9 7TF, UK; [157]Division of Cancer Epidemiology and Genetics, National Cancer Institute, National Institutes of Health, Bethesda, Maryland 20892, USA.

