## [Peer Review File · Nature Communications]

Referee #1 (Remarks to the Author):

Fang et al have characterized the multi-cancer risk locus on 5p15.33 and propose a biological mechanism for the association. 5p15.33 is an important and established risk locus so unravelling the mechanism would indeed be a significant contribution in the cancer field.

This is a revised version of an interesting high quality work providing multiple layers of evidence to support the claims.

1. The fine mapping results have been sufficiently improved.

We are pleased that the fine-mapping has been clarified and extended in the text to address this referee's concerns.

2. Annotations for rs3030832, rs27071 and rs27068 are missing from the top panel of Figure 1

Annotations for these three variants have now been added to Figure 1.

3. The ChIA-PET data highlighting in Figure 1 is still misleading. One either needs to remove the highlighting on the region 2 - TERT linking read or add the highlight also to the region 2 - CLPTM1L linking reads.

The red highlighting of the ChIA-PET in Figure 1 has been changed to grey so that it is not emphasized over other interactions in the region.

4. Similar to my first round comments; recombination hotspot likelihood ratios in Figure 1 and candlestick plots in Figures 4, 5 and 6 would be better with less processed data (recombination rate in Fig 1 and individual data points instead of candlesticks).

As requested, we have changed Figure 1 to include recombination rate. We have also included the recombination hotspots as we feel this provides additional information for this region.

Furthermore, as this reviewer suggested in both the first and current review to present individual data points instead of candlesticks for Figure 4, we have changed the format as requested for the allele-specific *TERT* inhibition after siRNA mediated inhibition of the enhancer encompassing rs36115365 (Figure 4c). We have similarly changed Figures 6a and 6c. However, we feel that visually this makes Figure 4b considerably less clear to a reader. Nonetheless, we have appended to this rebuttal an alternate version of Figure 4b with the data from independent experiments plotted individually, so that the reviewer and editor can see both versions. If needed, we are happy to include this new version of this figure. Figure 5e is now **Extended Data Figure 12g** showing individual data points from seven independent experiments (see below where the chromatin immunoprecipitation of ZNF148 over rs36115365 is addressed in response to questions from Reviewer #2).

The authors have also added the requested figure to supplement (SFigure 1). Unfortunately the base as well as the whole region appears to be poorly conserved. While this does not exclude a role in cancer one would have presumed that a key genetic locus involved in such a fundamental cellular process was conserved.

The region immediately surrounding rs36115365 does indeed only appear to be conserved in primates. While broader conservation may be expected for germline variants conferring a large

effect, we suggest that the lack of conservation beyond primates is not inconsistent with the small effect sizes observed for most cancer GWAS loci, including those observed within the *TERT-CLPTM1L* locus. Further, we stress that the region surrounding rs36115365 is not expected to be the only element regulating *TERT* in primates, consistent with both the small effect sizes observed for this locus as well as the presence of numerous independent cancer risk variants across the broader region harboring *TERT* and *CLPTM1L*.

Referee #2 (Remarks to the Author):

This paper addresses an important question regarding which sequences in the *TERT-CLPTM1L* region affect telomerase and what is the transcription factor that works through these sequences. The first half of this effort is on its own a major problem that needs answering and the authors use GWA approaches, siRNA approaches and luciferase assays to get at this part.

Determining which TF regulates the C allele is a daunting task and this resubmission falls short of showing that ZNF148 is the factor that works through this sequence. The data as to whether ZNF148 binds only the C or G allele - or alternatively binds both, but binds the C allele with higher affinity - are inconsistent. The mass spec pull downs and EMSA from extracts do not definitively establish ZNF148 as binding to these elements *in vivo* - they only establish the possibility that they bind *in vivo*. ChIP-seq is necessary to address this question. The ChIP PCR are preliminary as presented. Individual data for each allele are not shown and the subtle preference for the C over the G allele are seemingly at odds with the MS findings, and the nuclear extract EMSA and the recombinant protein EMSA.

We would like to emphasize that our *in vitro* results (EMSA and proteomics) support preferential rather than allele-specific binding of ZNF148 to rs36115365. In the EMSAs presented in the manuscript, the allelic banding observed using cell line nuclear extracts appears to be preferential rather than completely specific as shown by the fact that the G-allele competitor competes the band at higher doses on many occasions (**Figures 2 and 5b, Extended Data Figure 3**). Importantly, our mass spectrometry data is relatively quantitative in nature and can only support the conclusion of allele-preferential binding. The C/G ratios for specific dimethyl labelled ZNF148 peptides in our raw proteomic pulldown data range from 8.4-17.2 fold (average 14.0 fold). These ratios provide some measure of the differences in binding affinities between the two alleles and support the notion of allele preferred rather than allele specific binding of ZNF148 to the C allele at rs36115365. The raw proteomic data can be included in the manuscript if the editor and reviewers find this helpful. **We have changed the wording in the manuscript to clarify this point and to better reflect the preferential rather than specific nature of ZNF148 binding at rs36115365.**

To address the reviewer concerns related to ZNF148 binding to rs36115365 *in vivo*, we performed additional experiments by ChIP-PCR. First, we scanned the region surrounding rs36115365 using ChIP-PCR and observed enrichment of the ChIP signal for ZNF148 over the SNP as compared to the surrounding area (of up to 1 kb in each direction, see **Figure 5e and Extended Data Figure 12**) in multiple cell lines, **providing strong support for *in vivo* binding of ZNF148 at rs36115365**. We also performed additional ChIP-PCR experiments (in total seven independent immunoprecipitation experiments for each of two heterozygous cell lines) and do see a trend towards a greater pulldown of the C allele as compared to the G allele at rs36115365. This difference (1.12-1.51 fold), however, is only significant in one cell line (A549, $P=0.01$; Panc 05.04, $P=0.06$). These data are included as an Extended Data Figure (**Extended Data Figure 12g**) and are also addressed in the discussion. We note that we did also attempt to assay allelic enrichment using targeted ChIP-seq over rs36115365 multiple times as suggested by our previous editor at

Nature, but the experiments were not robust (small but variable number of reads generated and highly variable allele counts even within replicates generated from the same samples).

We do feel that our results are nonetheless consistent with a common susceptibility variant with a low effect on cancer risk over an individual's lifetime (ORs in the range of 1.15-1.30). Our EMSA and proteomics results are very stable and upon multiple repeated experiments in different cell lines, we consistently see more binding to the C allele than to the G allele. Although this is indeed an *in vitro* setting, the ChIP-PCR and ChIP-seq may not be sensitive enough to reliably detect the very subtle differences in the pulldown of the two alleles expected for a common susceptibility variant conferring a small effect on disease risk that is likely also to require additional cofactors. **We have added these data to the manuscript and discuss the suggestive findings.**

The telomerase assays are preliminary - there is no rescue for off target effects of the siRNAs and additional ZNF protein siRNAs (in addition to ZNF148) could have been tested.

To address this point in a manner previously suggested by the reviewer, we attempted to create stable CRISPR clones with inactivated *ZNF148* from A549 lung cancer cells. Unfortunately, after considerable screening of clones, the only clones we were able to obtain using this strategy retained protein that was both detectable using the ZNF148 antibody and bound preferentially to the C-allele of rs36115365 in EMSA assays. Sequencing of the CRISPR clones showed that all knock out events in these clones were heterozygous; no homozygous cell lines were isolated. Not all genes are amenable to stable CRISPR-based knock out, and our data are consistent with *ZNF148* being an essential gene. Consistent with this hypothesis, homozygous *ZNF148* (mouse gene name *Zfp148*) knock-out mice are embryonic lethal according to the International Mouse Phenotyping Consortium (IMPC); the phenotype is listed as 100% pre-weaning lethality (defined as lethality between fertilization and weaning age). Additionally, in the deCODE study of rare complete human knockouts (Sulem *et al.*, *Nature Genetics*, 2015; PMID: 25807282), *ZNF148* is not listed, further indicating that it may be essential.

As an alternative approach to address the potential for off-target effects of siRNAs targeting *ZNF148*, we instead knocked down endogenous *ZNF148* using an siRNA targeting the 3'-UTR, and have rescued the effects of siRNA-mediated knockdown by introducing exogenous *ZNF148*. Using this approach, we observe rescue of both *TERT* expression and telomerase activity; **these data are now included in the manuscript as Extended Data Figure 17.** We also note that in our last manuscript submission, we previously targeted *ZNF148* using four different siRNAs (now Extended Data Figure 14, previously Extended Data Figure 14B), as well as using siRNAs targeting the region surrounding rs36115365 rather than *ZNF148* itself, with nearly identical phenotypic effects, further arguing against these phenotypes being the result of off-target effects. We argue that these data should be sufficient to address this point.

We previously did perform siRNA knock down experiments for all three of the other zinc-finger proteins identified from our proteomic screen, *VEZF1* (ZNF161), *ZNF281*, and *ZNF740*, and observed no effect on *TERT* expression. **We have now included these results indicating that these three transcription factors do not regulate TERT. These data are described in the Results section (lines 250-252) and shown as Extended Data Figure 15.**

Telomeres cannot shorten with short term exposure to siRNAs because telomere shortening is a gradual process.

We chose a time-point of 20 days post-inhibition based on prior literature showing the effects of

inhibiting *TERT* or telomerase activity on telomere length. We provide below a few published and peer-reviewed examples of inhibition of *TERT* or telomerase activity on telomere length in a similar timeframe. **These data are consistent with our own results in Figure 6 that indicate that telomeres can indeed shorten with short term (measured in days to a few weeks) inhibition of *TERT* and *ZNF148* expression:**

de Souza Nascimento P *et al.*, *Oncol Rep*, 2006, 16(2):423-8.

shRNA knockdown reduced telomere length of HT29 cells after **10-20 population doublings**.

Zhou JM *et al.*, *Oncogene*, 2006, 25: 503-511.

Inhibiting telomerase activity shortened the telomeres of SW620 colon cancer cells in **18 days**.

Killedar A *et al.*, *PLoS Genet*, 2015, 11:6:e1005286.

Introduction of a partially dominant-negative splice variant of hTERT caused by rs10069690 (an independent cancer risk locus identified via GWAS) reduced telomere length in HT1080 cells in **35-50 population doublings**.

Referee #3 (Remarks to the Author):

The authors addressed adequately all of my comments. They also include new experiments in response to the other reviewer comments, all of which increase the confidence in their conclusions. I have no other revisions to suggest.

We thank the reviewer for his/her suggestions and positive comments.

Alternate Figure 4b

Reviewers' Comments:

Reviewer #1 (Remarks to the Author)

The paper has improved; still in Figure 1 one could omit the likelihood ratio (red) and let the recombination rate (blue) remain. Other than that I have no further issues.

Reviewer #2 (Remarks to the Author)

The authors have addressed all my previous concerns.

Response to Reviewer Comments
Fang *et al.*

We thank the reviewers and editors for careful consideration of our manuscript and their constructive comments that we feel have helped us improve the manuscript.

REVIEWERS' COMMENTS:

Reviewer #1 (Remarks to the Author):

The paper has improved; still in Figure 1 one could omit the likelihood ratio (red) and let the recombination rate (blue) remain. Other than that I have no further issues.

We thank the reviewer for his/her positive remarks. As per our discussions with the editor handling our manuscript at Nature Communications we are leaving both the likelihood ratio and recombination rate in Figure 1 as we feel both sets of data are informative for readers of the manuscript.

Reviewer #2 (Remarks to the Author):

The authors have addressed all my previous concerns.

We thank Reviewer 2 for her/his review of our manuscript.